# Global Non-convex Optimization
# with Discretized Diffusions

**Murat A. Erdogdu** [1,2]
erdogdu@cs.toronto.edu

**Lester Mackey** [3]
lmackey@microsoft.com

**Ohad Shamir** [4]
ohad.shamir@weizmann.ac.il

[1]University of Toronto  [2]Vector Institute  [3]Microsoft Research  [4]Weizmann Institute of Science

## Abstract

An Euler discretization of the Langevin diffusion is known to converge to the global minimizers of certain convex and non-convex optimization problems. We show that this property holds for any suitably smooth diffusion and that different diffusions are suitable for optimizing different classes of convex and non-convex functions. This allows us to design diffusions suitable for globally optimizing convex and non-convex functions not covered by the existing Langevin theory. Our non-asymptotic analysis delivers computable optimization and integration error bounds based on easily accessed properties of the objective and chosen diffusion. Central to our approach are new explicit Stein factor bounds on the solutions of Poisson equations. We complement these results with improved optimization guarantees for targets other than the standard Gibbs measure.

## 1   Introduction

Consider the unconstrained and possibly non-convex optimization problem

$$\underset{x \in \mathbb{R}^d}{\text{minimize}}\ f(x).$$

Recent studies have shown that the *Langevin algorithm* – in which an appropriately scaled isotropic Gaussian vector is added to a gradient descent update – globally optimizes $f$ whenever the objective is *dissipative* ($\langle \nabla f(x), x \rangle \geq \alpha \|x\|_2^2 - \beta$ for $\alpha > 0$) with a Lipschitz gradient [14, 25, 29]. Remarkably, these globally optimized objectives need not be convex and can even be multimodal. The intuition behind the success of the Langevin algorithm is that the stochastic optimization method approximately tracks the continuous-time Langevin diffusion which admits the *Gibbs measure* – a distribution defined by $p_\gamma(x) \propto \exp(-\gamma f(x))$ – as its invariant distribution. Here, $\gamma > 0$ is an inverse temperature parameter, and when $\gamma$ is large, the Gibbs measure concentrates around its modes. As a result, for large values of $\gamma$, a rapidly mixing Langevin algorithm will be close to a global minimum of $f$. In this case, rapid mixing is ensured by the Lipschitz gradient and dissipativity. Due to its simplicity, efficiency, and well-understood theoretical properties, the Langevin algorithm and its derivatives have found numerous applications in machine learning [see, e.g., 28, 6].

In this paper, we prove an analogous global optimization property for the Euler discretization of any smooth and dissipative diffusion and show that different diffusions are suitable for solving different classes of convex and non-convex problems. Our non-asymptotic analysis, based on a multidimensional version of Stein's method, establishes explicit bounds on both integration and optimization error. Our contributions can be summarized as follows:

- For any function $f$, we provide explicit $\mathcal{O}\!\left(\frac{1}{\epsilon^2}\right)$ bounds on the numerical integration error of discretized dissipative diffusions. Our bounds depend only on simple properties of the

diffusion's coefficients and Stein factors, i.e., bounds on the derivatives of the associated Poisson equation solution.

- For pseudo-Lipschitz $f$, we derive explicit first through fourth-order Stein factor bounds for every fast-coupling diffusion with smooth coefficients. Since our bounds depend on Wasserstein coupling rates, we provide user-friendly, broadly applicable tools for computing these rates. The resulting computable integration error bounds recover the known Markov chain Monte Carlo convergence rates of the Langevin algorithm in both convex and non-convex settings but apply more broadly.

- We introduce new explicit bounds on the expected suboptimality of sampling from a diffusion. Together with our integration error bounds, these yield computable and convergent bounds on global optimization error. We demonstrate that improved optimization guarantees can be obtained by targeting distributions other than the standard Gibbs measure.

- We show that different diffusions are appropriate for different objectives $f$ and detail concrete examples of global non-convex optimization enabled by our framework but not covered by the existing Langevin theory. For example, while the Langevin diffusion is particularly appropriate for dissipative and hence quadratic growth $f$ [25, 29], we show alternative diffusions are appropriate for "heavy-tailed" $f$ with subquadratic or sublinear growth.

We emphasize that, while past work has assumed the existence of finite Stein factors [4, 29], focused on deriving convergence rates with inexplicit constants [23, 26, 29], or concentrated singularly on the Langevin diffusion [6, 9, 29, 25], the goals of this work are to provide the reader with tools to (a) check the appropriateness of a given diffusion for optimizing a given objective and (b) compute explicit optimization and integration error bounds based on easily accessed properties of the objective and chosen diffusion. The rest of the paper is organized as follows. Section 1.1 surveys related work. Section 2 provides an introduction to diffusions and their use in optimization and reviews our notation. Section 3 provides explicit bounds on integration error in terms of Stein factors and on Stein factors in terms of simple properties of $f$ and the diffusion. In Section 4, we provide explicit bounds on optimization error by targeting Gibbs and non-Gibbs invariant measures and discuss how to obtain better optimization error using non-Gibbs invariant measures. We give concrete examples of applying these tools to non-convex optimization problems in Section 5 and conclude in Section 6.

## 1.1 Related work

The Euler discretization of the Langevin diffusion is commonly termed the Langevin algorithm and has been studied extensively in the context of sampling from a log concave distribution. Non-asymptotic integration error bounds for the Langevin algorithm are studied in [8, 7, 9, 10]. A representative bound follows from combining the ergodicity of the diffusion with a discretization error analysis and yields $\epsilon$ error in $\mathcal{O}(\frac{1}{\epsilon^2}\mathrm{poly}(\log(\frac{1}{\epsilon})))$ steps for the strongly log concave case and $\mathcal{O}(\frac{1}{\epsilon^4}\mathrm{poly}(\log(\frac{1}{\epsilon})))$ steps for the general log concave case [7, 9].

Our work is motivated by a line of research that uses the Langevin algorithm to globally optimize non-convex functions. Gelfand and Mitter [14] established the global convergence of an appropriate variant of the algorithm, and Raginsky et al. [25] subsequently used optimal transport theory to prove optimization and integration error bounds. For example, [25] provides an integration error bound of $\epsilon$ after $\mathcal{O}\left(\frac{1}{\epsilon^4}\mathrm{poly}(\log(\frac{1}{\epsilon}))\frac{1}{\lambda_*}\right)$ steps under the quadratic-growth assumptions of dissipativity and a Lipschitz gradient; the estimate involves the inverse spectral gap parameter $\lambda_*^{-1}$, a quantity that is often unknown and sometimes exponential in both inverse temperature and dimension. Gao et al. [13] obtained similar guarantees for stochastic Hamiltonian Monte Carlo algorithms for empirical and population risk minimization under a dissipativity assumption with rate estimates. In this work, we accommodate "heavy-tailed" objectives that grow subquadratically and trade the often unknown and hence inexplicit spectral gap parameter of [25] for the more user-friendly distant dissipativity condition (Prop. 3.4) which provides a straightforward and explicit certification of fast coupling and hence the fast mixing of a diffusion. For distantly dissipative diffusions, the size of our error bounds is driven primarily by a computable distance parameter; in the Langevin setting, an analogous quantity is studied in place of the spectral gap in the contemporaneous work of [5].

Cheng et al. [5] provide integration error bounds for sampling with the overdamped Langevin algorithm under a distant strong convexity assumption (a special case of distant dissipativity). The authors build on the results of [9, 11] and establish $\epsilon$ error in $\mathcal{O}(\frac{1}{\epsilon^2}\log(\frac{1}{\epsilon}))$ steps. We consider general

distantly dissipative diffusions and establish an integration error of $\epsilon$ in $\mathcal{O}(\frac{1}{\epsilon^2})$ steps under mild assumptions on the objective function $f$ and smoothness of the diffusion.

Vollmer et al. [26] used the solution of the Poisson equation in their analysis of stochastic Langevin gradient descent, invoking the bounds of Pardoux and Veretennikov [24, Thms. 1 and 2] to obtain Stein factors. However, Thms. 1 and 2 of [24] yield only inexplicit constants and require bounded diffusion coefficients, a strong assumption violated by the examples treated in Section 5. Chen et al. [4] considered a broader range of diffusions but assumed, without verification, that Stein factor and Markov chain moment were universally bounded by constants independent of all problem parameters. One of our principal contributions is a careful enumeration of the dependencies of these Stein factors and Markov chain moments on the objective $f$ and the candidate diffusion. Our convergence analysis builds on the arguments of [23, 15], and our Stein factor bounds rely on distant and uniform dissipativity conditions for $L_1$-Wasserstein rate decay [11, 15] and the smoothing effect of the Markov semigroup [3, 15]. Our Stein factor results significantly generalize the existing bounds of [15] by accommodating pseudo-Lipschitz objectives $f$ and quadratic growth in the covariance coefficient and deriving the first four Stein factors explicitly.

## 2 Optimization with Discretized Diffusions: Preliminaries

Consider a target objective function $f : \mathbb{R}^d \to \mathbb{R}$. Our goal is to carry out unconstrained minimization of $f$ with the aid of a candidate diffusion defined by the stochastic differential equation (SDE)

$$dZ_t^z = b(Z_t^z)dt + \sigma(Z_t^z)dB_t \quad \text{with} \quad Z_0^z = z. \tag{2.1}$$

Here, $(B_t)_{t \geq 0}$ is an $l$-dimensional Wiener process, and $b : \mathbb{R}^d \to \mathbb{R}^d$ and $\sigma : \mathbb{R}^d \to \mathbb{R}^{d \times l}$ represent the drift and the diffusion coefficients, respectively. The diffusion $Z_t^z$ starts at a point $z \in \mathbb{R}^d$ and, under the conditions of Section 3, admits a limiting invariant distribution $P$ with (Lebesgue) density $p$. To encourage sampling near the minima of $f$, we would like to choose $p$ so that the maximizers of $p$ correspond to minimizers of $f$. Fortunately, under mild conditions, one can construct a diffusion with target invariant distribution $P$ (see, e.g., [20, 15, Thm. 2]), by selecting the drift coefficient

$$b(x) = \frac{1}{2p(x)}\langle \nabla, p(x)(a(x) + c(x)) \rangle, \tag{2.2}$$

where $a(x) \triangleq \sigma(x)\sigma(x)^\top$ is the covariance coefficient, $c(x) = -c(x)^\top \in \mathbb{R}^{d \times d}$ is the skew-symmetric stream coefficient, and $\langle \nabla, m(x) \rangle = \sum_j e_j \sum_k \frac{\partial m_{jk}(x)}{\partial x_k}$ denotes the divergence operator with $\{e_j\}_j$ as the standard basis of $\mathbb{R}^d$. As an illustration, consider the (overdamped) Langevin diffusion for the Gibbs measure with inverse temperature $\gamma > 0$ and density

$$p_\gamma(x) \propto \exp(-\gamma f(x)) \tag{2.3}$$

associated with our objective $f$. Inserting $\sigma(x) = \sqrt{2/\gamma}\, I$ and $c(x) = 0$ into the formula (2.2) we obtain

$$b_j(x) = \frac{1}{2p_\gamma(x)}\langle \nabla, p_\gamma(x)(a(x) + c(x)) \rangle_j = \frac{1}{\gamma p_\gamma(x)}\sum_k \frac{\partial p_\gamma(x) I_{jk}}{\partial x_k} = \frac{1}{\gamma p_\gamma(x)}\frac{\partial p_\gamma(x)}{\partial x_j} = -\frac{\partial_j f(x)}{\partial x_j},$$

which reduces to $b = -\nabla f$. We emphasize that the choice of the Gibbs measure is arbitrary, and we will consider other measures that yield superior guarantees for certain minimization problems.

In practice, the diffusion (2.1) cannot be simulated in continuous time and is instead approximated by a discrete-time numerical integrator. We will show that a particular discretization, the Euler method, can be used as a global optimization algorithm for various families of convex and non-convex $f$. The Euler method is the most commonly used discretization technique due to its explicit form and simplicity; however, our analysis can be generalized to other numerical integrators as well. For $m = 0, 1, ...,$ the Euler discretization of the SDE (2.1) corresponds to the Markov chain updates

$$X_{m+1} = X_m + \eta\, b(X_m) + \sqrt{\eta}\, \sigma(X_m)W_m,$$

where $\eta$ is the step size, and $W_m \sim \mathsf{N}_d(0, I)$ is an isotropic Gaussian vector that is independent from $X_m$. This update rule defines a Markov chain which typically has an invariant measure that is different from the invariant measure of the continuous time diffusion. However, when the step size $\eta$ is sufficiently small, the difference between two invariant measures becomes small and can be quantitatively characterized [see, e.g., 22]. Our optimization algorithm is simply to evaluate the function $f$ at each Markov chain iterate $X_m$ and report the point with the smallest function value.

Denoting by $p(f)$ the expectation of $f$ under the density $p$ – i.e., $p(f) = \mathbb{E}_{Z \sim p}[f(Z)]$ – we decompose the optimization error after $M$ steps of our Markov chain into two components,

$$\min_{m=1,..,M} \mathbb{E}[f(X_m)] - \min_x f(x) \leq \underbrace{\tfrac{1}{M} \sum_{m=1}^{M} \mathbb{E}[f(X_m) - p(f)]}_{\text{integration error}} + \underbrace{p(f) - \min_x f(x)}_{\text{expected suboptimality}}, \quad (2.4)$$

and bound each term on the right-hand side separately. The integration error—which captures both the short-term non-stationarity of the chain and the long-term bias due to discretization—is the subject of Section 3; we develop explicit bounds using techniques that build upon [23, 15]. The expected suboptimality quantifies how well exact samples from $p$ minimize $f$ on average. In Section 4, we extend the Gibbs measure Langevin diffusion bound of Raginsky et al. [25] to more general invariant measures and associated diffusions and demonstrate the benefits of targeting non-Gibbs measures.

**Notation.** We say a function $g$ is pseudo-Lipschitz continuous of order $n$ if it satisfies

$$|g(x) - g(y)| \leq \tilde{\mu}_{1,n}(g)(1 + \|x\|_2^n + \|y\|_2^n)\|x - y\|_2, \quad \text{for all } x, y \in \mathbb{R}^d, \quad (2.5)$$

where $\|\cdot\|_2$ denotes the Euclidean norm, and $\tilde{\mu}_{1,n}(g)$ is the smallest constant satisfying (2.5). This assumption, which relaxes the more stringent Lipschitz assumption, allows $g$ to exhibit polynomial growth of order $n$. For example, $g(x) = x^2$ is not Lipschitz but satisfies (2.5) with $\tilde{\mu}_{1,1}(g) \leq 1$. In all of our examples of interest, $n \leq 1$. For operator and Frobenius norms $\|\cdot\|_{\text{op}}$ and $\|\cdot\|_{\text{F}}$, we use

$$\phi_1(g) = \sup_{x,y \in \mathbb{R}^d, x \neq y} \tfrac{\|g(x) - g(y)\|_{\text{F}}}{\|x - y\|_2}, \qquad \mu_0(g) = \sup_{x \in \mathbb{R}^d} \|g(x)\|_{\text{op}},$$

$$\text{and} \quad \mu_i(g) = \sup_{x,y \in \mathbb{R}^d, x \neq y} \tfrac{\|\nabla^{i-1} g(x) - \nabla^{i-1} g(y)\|_{\text{op}}}{\|x - y\|_2}$$

for the $i$-th order Lipschitz coefficients of a sufficiently differentiable function $g$. We denote the degree $n$ polynomial coefficient of the $i$-th derivative of $g$ by $\tilde{\pi}_{i,n}(g) \triangleq \sup_{x \in \mathbb{R}^d} \tfrac{\|\nabla^i g(x)\|_{\text{op}}}{1 + \|x\|_2^n}$.

## 3 Explicit Bounds on Integration Error

We develop our explicit bounds on integration error in three steps. In Theorem 3.1, we bound integration error in terms of the polynomial growth and dissipativity of diffusion coefficients (Conditions 1 and 2) and Stein factors bounds on the derivatives of solutions to the diffusion's Poisson equation (Condition 3). Condition 3 is a common assumption in the literature but is typically not verified. To address this shortcoming, Theorem 3.2 shows that any smooth, fast-coupling diffusion admits finite Stein factors expressed in terms of diffusion coupling rates (Condition 4). Finally, in Section 3.1, we provide user-friendly tools for explicitly bounding those diffusion coupling rates. We begin with our conditions.

**Condition 1** (Polynomial growth of coefficients). *For some $r \in \{1, 2\}$ and $\forall x \in \mathbb{R}^d$, the drift and the diffusion coefficients of the diffusion (2.1) satisfy the growth condition*

$$\|b(x)\|_2 \leq \tfrac{\lambda_b}{4}(1 + \|x\|_2), \quad \|\sigma(x)\|_{\text{F}} \leq \tfrac{\lambda_\sigma}{4}(1 + \|x\|_2), \quad \text{and} \quad \|\sigma\sigma^\top(x)\|_{\text{op}} \leq \tfrac{\lambda_a}{4}(1 + \|x\|_2^r).$$

The existence and uniqueness of the solution to the diffusion SDE (2.1) is guaranteed under Condition 1 [19, Thm 3.5]. The cases $r = 1$ and $r = 2$ correspond to linear and quadratic growth of $\|\sigma\sigma^\top(x)\|_{\text{op}}$, and we will explore examples of both $r$ settings in Section 5. As we will see in each result to follow, the quadratic growth case is far more delicate.

**Condition 2** (Dissipativity). *For $\alpha, \beta > 0$, the diffusion (2.1) satisfies the dissipativity condition*

$$\mathcal{A}\|x\|_2^2 \leq -\alpha\|x\|_2^2 + \beta \quad \text{for} \quad \mathcal{A}g(x) \triangleq \langle b(x), \nabla g(x) \rangle + \tfrac{1}{2}\langle \sigma(x)\sigma(x)^\top, \nabla^2 g(x) \rangle. \quad (3.1)$$

*$\mathcal{A}$ is the generator of the diffusion with coefficients $b$ and $\sigma$, and $\mathcal{A}\|x\|_2^2 = 2\langle b(x), x \rangle + \|\sigma(x)\|_{\text{F}}^2$.*

Dissipativity is a standard assumption that ensures that the diffusion does not diverge but rather travels inward when far from the origin [22]. Notably, a linear growth bound on $\|\sigma(x)\|_{\text{F}}$ and a quadratic growth bound on $\|\sigma\sigma^\top(x)\|_{\text{op}}$ follow directly from the linear growth of $\|b(x)\|$ and Condition 2. However, in many examples, tighter growth constants can be obtained by inspection.

Our final condition concerns the solution of the Poisson equation (also known as the Stein equation in the Stein's method literature) associated with our candidate diffusion.

**Condition 3** (Finite Stein factors). *The function $u_f$ solves the Poisson equation with generator* (3.1)

$$f - p(f) = \mathcal{A}u_f, \tag{3.2}$$

*is pseudo-Lipschitz of order $n$ with constant $\zeta_1$, and has $i$-th order derivative with degree-$n$ polynomial growth for $i = 2, 3, 4$, i.e.,*

$$\|\nabla^i u_f(x)\|_{\mathrm{op}} \le \zeta_i(1 + \|x\|_2^n) \text{ for } i \in \{2, 3, 4\} \text{ and all } x \in \mathbb{R}^d.$$

*In other words, $\tilde{\mu}_{1,n}(u_f) = \zeta_1$, and $\tilde{\pi}_{i,n}(u_f) = \zeta_i$ for $i = 2, 3, 4$ with $\max_i \zeta_i < \infty$.*

The coefficients $\zeta_i$ govern the regularity of the Poisson equation solution $u_f$ and are termed Stein factors in the Stein's method literature. Although variants of Condition 3 have been assumed in previous work [4, 26], we emphasize that this assumption is not easily verified, and frequently only empirical evidence is provided as justification for the assumption [4]. We will ultimately derive explicit expressions for the Stein factors $\zeta_i$ for a wide variety of diffusions and functions $f$, but first we will use the Stein factors to bound the integration error of our discretized diffusion.

**Theorem 3.1** (Integration error of discretized diffusions). *Let Conditions 1 to 3 hold for some $r \in \{1, 2\}$. For any even integer[1] $n_e \ge n + 4$ and a step size satisfying $\eta < 1 \wedge \frac{\alpha}{2(n_e - 1)!!(1 + \lambda_b/2 + \lambda_\sigma/2)^{n_e}}$,*

$$\left| \frac{1}{M} \sum_{m=1}^{M} \mathbb{E}[f(X_m)] - p(f) \right| \le \left( c_1 \frac{1}{\eta M} + c_2 \eta + c_3 \eta^{1 + |1 \wedge n/2|} \right) \left( \kappa_r(n_e) + \mathbb{E}[\|X_0\|_2^{n_e}] \right),$$

*where*

$$c_1 = 6\zeta_1, \qquad c_2 = \frac{1}{16}\left[ 2\zeta_2 \lambda_b^2 + \zeta_3 \lambda_b \lambda_\sigma^2 + \zeta_4(1 + 3^{n-1})\lambda_\sigma^4 \right],$$

$$c_3 = \frac{1}{48}\left[ \zeta_3 \lambda_b^3 + \zeta_4 \lambda_b^4(1 + 3^{n-1}) + 4\zeta_4 \frac{1.5^n}{n^4}(\lambda_b^4 + n_e^2 \lambda_\sigma^4)(\lambda_b^n + n!!\lambda_\sigma^n) \right],$$

$$\kappa_r(n) = 2 + \frac{2\beta}{\alpha} + \frac{n\lambda_a}{4\alpha} + \frac{\tilde{\alpha}_r}{\alpha}\left( \frac{n\lambda_a + 6r\beta}{2r\tilde{\alpha}_r} \right)^n, \quad \text{with} \quad \tilde{\alpha}_1 = \alpha, \; \tilde{\alpha}_2 = [\alpha - n_e \lambda_a/4]_+.$$

This integration error bound, proved in Appendix A, is $\mathcal{O}\left( \frac{1}{\eta M} + \eta \right)$ since the higher order term $c_3 \eta^{1 + |1 \wedge n/2|}$ can be combined with the dominant term $c_2 \eta$ yielding $(c_2 + c_3)\eta$ as $\eta < 1$. We observe that one needs $\mathcal{O}(\epsilon^{-2})$ steps to reach a tolerance of $\epsilon$. Theorem 3.1 seemingly makes no assumptions on the objective function $f$, but in fact the dependence on $f$ is present in the growth parameters, the Stein factors, and the polynomial degree of the Poisson equation solution. For example, we will show in Theorem 3.2 that the polynomial degree is upper bounded by that of the objective function $f$. To characterize the function classes covered by Theorem 3.1, we next turn to dissecting the Stein factors.

While verifying Conditions 1 and 2 for a given diffusion is often straightforward, it is not immediately clear how one might verify Condition 3. As our second principal contribution, we derive explicit values for the Stein factors $\zeta_i$ for any smooth and dissipative diffusion exhibiting fast $L_1$-Wasserstein decay:

**Condition 4** (Wasserstein rate). *The diffusion $Z_t^x$ has $L_p$-Wasserstein rate $\varrho_p : \mathbb{R}_{\ge 0} \to \mathbb{R}$ if*

$$\inf\nolimits_{\text{couplings } (Z_t^x, Z_t^y)} \mathbb{E}[\|Z_t^x - Z_t^y\|_2^p]^{1/p} \le \varrho_p(t)\|x - y\|_2 \quad \text{for all } x, y \in \mathbb{R}^d \text{ and } t \ge 0,$$

*where infimum is taken over all couplings between $Z_t^x$ and $Z_t^y$. We further define the relative rates*

$$\tilde{\varrho}_1(t) = \log(\varrho_2(t)/\varrho_1(t)) \quad \text{and} \quad \tilde{\varrho}_2(t) = \log(\varrho_1(t)/[\varrho_1(0)\varrho_2(t)])/\log(\varrho_1(t)/\varrho_1(0)).$$

**Theorem 3.2** (Finite Stein factors from Wasserstein decay). *Assume that Conditions 1, 2 and 4 hold and that $f$ is pseudo-Lipschitz continuous of order $n$ with, for $i = 2, 3, 4$, at most degree-$n$ polynomial growth of its $i$-th order derivatives. Then, Condition 3 is satisfied with Stein factors*

$$\zeta_i = \tau_i + \xi_i \int_0^\infty \varrho_1(t)\omega_r(t + i - 2)dt \quad \text{for} \quad i = 1, 2, 3, 4, \quad \text{where}$$

$$\omega_r(t) = 1 + 4\varrho_1(t)^{1-1/r}\varrho_1(0)^{1/2}\left( 1 + \frac{2}{\tilde{\alpha}_r^n}\{[1 \vee \tilde{\varrho}_r(t)]2\lambda_a n + 3r\beta\}^n \right),$$

*with $\tilde{\alpha}_1 = \alpha$, $\tilde{\alpha}_2 = \inf_{t \ge 0}[\alpha - n\lambda_a(1 \vee \tilde{\varrho}_2(t))]_+$, and*

$$\tau_1 = 0 \qquad \& \quad \tau_i = \tilde{\mu}_{1,n}(f)\tilde{\pi}_{2:i,n}(f)\tilde{\nu}_{1:i}(b)\tilde{\nu}_{1:i}(\sigma)\kappa_r(6n) \qquad \qquad \text{for} \quad i = 2, 3, 4,$$

$$\xi_1 = \tilde{\mu}_{1,n}(f) \quad \& \quad \xi_i = \tilde{\mu}_{1,n}(f)\tilde{\nu}_{1:i}(b)\tilde{\nu}_{1:i}(\sigma)\tilde{\nu}_{0:i-2}(\sigma^{-1})\varrho_1(0)\omega_r(1)\kappa_r(6n)^{i-1} \quad \text{for} \quad i = 2, 3, 4,$$

*where $\kappa_r(n)$ is as in Theorem 3.1, $\tilde{\pi}_{a:b,n}(f) = \max_{i=a,..,b} \tilde{\pi}_{i,n}(f)$, and $\tilde{\nu}_{a:b}(g)$ is a constant, given explicitly in the proof, depending only on the order $a$ through $b$ derivatives of $g$.*

The proof of Theorem 3.2 is given in Section B and relies on the explicit transition semigroup derivative bounds of [12]. We emphasize that, to provide finite Stein factors, Theorem 3.2 only requires $L_1$-Wasserstein decay and allows the $L_2$-Wasserstein rate to grow. An integrable Wasserstein rate is an indication that a diffusion mixes quickly to its stationary distribution. Hence, Theorem 3.2 suggests that, for a given $f$, one should select a diffusion that mixes quickly to a stationary measure that, like the Gibbs measure (2.3), has modes at the minimizers of $f$. We explore user-friendly conditions implying fast Wasserstein decay in Section 3.1 and detailed examples deploying these tools in Section 5. Crucially for the "heavy-tailed" examples given in Section 5, Theorem 3.2 allows for an unbounded diffusion coefficient $\sigma$, unlike the classic results of [24].

## 3.1 Sufficient conditions for Wasserstein decay

A simple condition that leads to exponential $L^1$ and $L^2$-Wasserstein decay is *uniform dissipativity* (3.3). The next result from [27] (see also [2, Sec. 1], [15, Thm. 10]) makes the relationship precise.

**Proposition 3.3** (Wasserstein decay from uniform dissipativity [27, Thm. 2.5]). *A diffusion with drift and diffusion coefficients $b$ and $\sigma$ has Wasserstein rate $\varrho_p(t) = e^{-kt/2}$ if, for all $x, y \in \mathbb{R}^d$,*

$$2\langle b(x) - b(y), x - y \rangle + \|\sigma(x) - \sigma(y)\|_F^2 + (p - 2)\|\sigma(x) - \sigma(y)\|_{op}^2 \leq -k\|x - y\|_2^2. \quad (3.3)$$

In the Gibbs measure Langevin case, where $b = -\nabla f$ and $\sigma \equiv \sqrt{2/\gamma} I$, uniform dissipativity is equivalent to the strong convexity of $f$. As we will see in Section 5, the extra degree of freedom in the diffusion coefficient $\sigma$ will allow us to treat non-convex and non-strongly convex functions $f$.

A more general condition leading to exponential $L_1$-Wasserstein decay is the *distant dissipativity* condition (3.4). The following result of [15] builds upon the pioneering analyses of Eberle [11, Cor. 2] and Wang [27, Thm. 2.6] to provide explicit Wasserstein decay.

**Proposition 3.4** (Wasserstein decay from distant dissipativity [15, Cor. 4.2]). *A diffusion with drift and diffusion coefficients $b$ and $\sigma$ satisfying $\tilde{\sigma}(x) \triangleq (\sigma(x)\sigma(x)^\top - s^2 I)^{1/2}$ and*

$$\frac{\langle b(x) - b(y), x - y \rangle}{s^2 \|x - y\|_2^2 / 2} + \frac{\|\tilde{\sigma}(x) - \tilde{\sigma}(y)\|_F^2}{s^2 \|x - y\|_2^2} - \frac{\|(\tilde{\sigma}(x) - \tilde{\sigma}(y))^\top (x - y)\|_2^2}{s^2 \|x - y\|_2^4} \leq \begin{cases} -K \text{ if } \|x - y\|_2 > R \\ L \text{ if } \|x - y\|_2 \leq R \end{cases} \quad (3.4)$$

*for $R, L \geq 0$, $K > 0$, and $s \in (0, 1/\mu_0(\sigma^{-1}))$ has Wasserstein rate $\varrho_1(t) = 2e^{LR^2/8}e^{-kt/2}$ for*

$$s^2 k^{-1} \leq \begin{cases} \frac{e-1}{2}R^2 + e\sqrt{8K^{-1}}R + 4K^{-1} & \text{if } LR^2 \leq 8 \\ 8\sqrt{2\pi}R^{-1}L^{-1/2}(L^{-1} + K^{-1})\exp(\frac{LR^2}{8}) + 32R^{-2}K^{-2} & \text{if } LR^2 > 8. \end{cases}$$

Conveniently, both uniform and distant dissipativity imply our dissipativity condition, Condition 2. The Prop. 3.4 rates feature the distance-dependent parameter $e^{LR^2/8}$. In the pre-conditioned Langevin Gibbs setting ($b = -\frac{1}{2}a\nabla f$ and $\sigma$ constant) when $f$ is the negative log likelihood of a multimodal Gaussian mixture, $R$ in (3.4) represents the maximum distance between modes [15]. When $R$ is relatively small, the convergence of the diffusion towards its stationary distribution is rapid, and the non-uniformity parameter is small; when $R$ is relatively large, the parameter grows exponentially in $R^2$, as would be expected due to infrequent diffusion transitions between modes.

Our next result, proved in Appendix D, provides a user-friendly set of sufficient conditions for verifying distant dissipativity and hence exponential Wasserstein decay in practice.

**Proposition 3.5** (User-friendly Wasserstein decay). *Fix any diffusion and skew-symmetric stream coefficients $\sigma$ and $c$ satisfying $L^* \triangleq F_1(\tilde{\sigma})^2 + \sup_x \lambda_{\max}(\nabla\langle\nabla, m(x)\rangle) < \infty$ for $m(x) \triangleq \sigma(x)\sigma(x)^\top + c(x)$, $\tilde{\sigma}(x) \triangleq (\sigma(x)\sigma(x)^\top - s_0^2 I)^{1/2}$, and $s_0 \in (0, 1/\mu_0(\sigma^{-1}))$. If*

$$\frac{-\langle m(x)\nabla f(x) - m(y)\nabla f(y), x - y \rangle}{\|x - y\|_2^2} \leq \begin{cases} -K_m & \text{if } \|x - y\|_2 > R_m \\ L_m & \text{if } \|x - y\|_2 \leq R_m, \end{cases} \quad (3.5)$$

*holds for $R_m, L_m \geq 0$, $K_m > 0$, then, for any inverse temperature $\gamma > L^*/K_m$, the diffusion with drift and diffusion coefficients $b_\gamma = -\frac{1}{2}m\nabla f + \frac{1}{2\gamma}\langle\nabla, m\rangle$ and $\sigma_\gamma = \frac{1}{\sqrt{\gamma}}\sigma$ has stationary density $p_\gamma(x) \propto e^{-\gamma f(x)}$ and satisfies (3.4) with $s = \frac{s_0}{\sqrt{\gamma}}$, $K = \frac{\gamma K_m - L^*}{s_0^2}$, $L = \frac{\gamma L_m + L^*}{s_0^2}$, and $R = R_m$.*

# 4 Explicit Bounds on Optimization Error

To convert our integration error bounds into bounds on optimization error, we now turn our attention to bounding the expected suboptimality term of (2.4). To characterize the expected suboptimality of sampling from a measure with modes matching the minima of $f$, we generalize a result due to Raginsky et al. [25]. The original result [25, Prop. 3.4] was designed to analyze the Gibbs measure (2.3) and demanded that $\log p_\gamma$ be smooth, in the sense that $\mu_2(\log p_\gamma) < \infty$. Our next proposition, proved in Appendix C, is designed for more general measures $p$ and importantly relaxes the smoothness requirements on $\log p$.

**Proposition 4.1** (Expected suboptimality: Sampling yields near-optima). *Suppose $p$ is the stationary density of an $(\alpha, \beta)$-dissipative diffusion (Condition 2) with global maximizer $x^*$. If $p$ takes the generalized Gibbs form $p_{\gamma,\theta}(x) \propto \exp(-\gamma(f(x) - f(x^*))^\theta)$ for $\gamma > 0$ and $\nabla f(x^*) = 0$, we have*

$$p_{\gamma,\theta}(f) - f(x^*) \ \leq \ \sqrt[\theta]{\frac{d}{2\gamma}(\frac{1}{\theta}\log(\frac{2\gamma}{d}) + \log(\frac{e\beta\mu_2(f)}{2\alpha}))}. \tag{4.1}$$

*More generally, if $\log p(x^*) - \log p(x) \leq C\|x - x^*\|_2^{2\theta}$ for some $C > 0$ and $\theta \in (0, 1]$ and all $x$, then*

$$-p(\log p) + \log p(x^*) \ \leq \ \frac{d}{2\theta}\log(\frac{2C}{d}) + \frac{d}{2}\log(\frac{e\beta}{\alpha}). \tag{4.2}$$

When $\theta = 1$, $p_{\gamma,\theta}$ is the Gibbs measure, and the bound (4.1) exactly recovers [25, Prop. 3.4]. The generalized Gibbs measures with $\theta < 1$ allow for improved dependence on the inverse temperature when $\gamma \gg d/(2\theta)$. Note however that, for $\theta < 1$, the distributions $p_{\gamma,\theta}$ also require knowledge of the optimal value $f(x^*)$. In certain practical settings, such as neural network optimization, it is common to have $f(x^*) = 0$. When $f(x^*)$ is unknown, a similar analysis can be carried out by replacing $f(x^*)$ with an estimate, and the bound (4.1) still holds up to a controllable error factor.

By combining Prop. 4.1 with Theorem 3.1, we obtain a complete bound controlling the global optimization error of the best Markov chain iterate.

**Corollary 4.2** (Optimization error of discretized diffusions). *Instantiate the assumptions and notation of Theorem 3.1 and Prop. 4.1. If the diffusion has the generalized Gibbs stationary density $p_{\gamma,\theta}(x) \propto \exp(-\gamma(f(x) - f(x^*))^\theta)$, then*

$$\min_{m=1,..,M} \mathbb{E}[f(X_m)] - f(x^*) \leq \Big(c_1\frac{1}{\eta M} + (c_2+c_3)\eta\Big)\Big(\kappa_r(n_e) + \mathbb{E}[\|X_0\|_2^{n_e}]\Big) \tag{4.3}$$
$$+ \sqrt[\theta]{\frac{d}{2\gamma}(\frac{1}{\theta}\log(\frac{2\gamma}{d}) + \log(\frac{e\beta\mu_2(f)}{2\alpha}))}.$$

Finally, we demonstrate that, for quadratic functions, the generalized Gibbs expected suboptimality bound (4.1) can be further refined to remove the $\log(\gamma/d)^{1/\theta}$ dependence.

**Proposition 4.3** (Expected suboptimality: Quadratic $f$). *Let $f(x) = \langle x - b, A(x - b)\rangle$ for a positive semidefinite $A \in \mathbb{R}^{d\times d}$ and $b \in \mathbb{R}^d$. Then for $p_{\gamma,\theta}(x) \propto \exp(-\gamma(f(x) - f(x^*))^\theta)$ with $\theta > 0$, and for each positive integer $k$, we have*

$$p_{\gamma,1/k}(f) - f(x^*) \ \leq \ \Big(\frac{k(1+\frac{d}{2})-1}{\gamma}\Big)^k. \tag{4.4}$$

The bound (4.4) applies to any $f$ with level set (i.e., $\{x : f(x) = \rho\}$) volume proportional to $\rho^{d-1}$.

# 5 Applications to Non-convex Optimization

We next provide detailed examples of verifying that a given diffusion is appropriate for optimizing a given objective, using either uniform dissipativity (Prop. 3.3) or our user-friendly distant dissipativity conditions (Prop. 3.5). When the Gibbs measure Langevin diffusion is used, our results yield global optimization when $f$ is strongly convex (condition (3.3) with $b = -\nabla f$ and $\sigma \equiv \sqrt{2/\gamma}I$) or has strongly convex tails (condition (3.5) with $m \equiv I$). To highlight the value of non-constant diffusion coefficients, we will focus on "heavy-tailed" examples that are not covered by the Langevin theory.

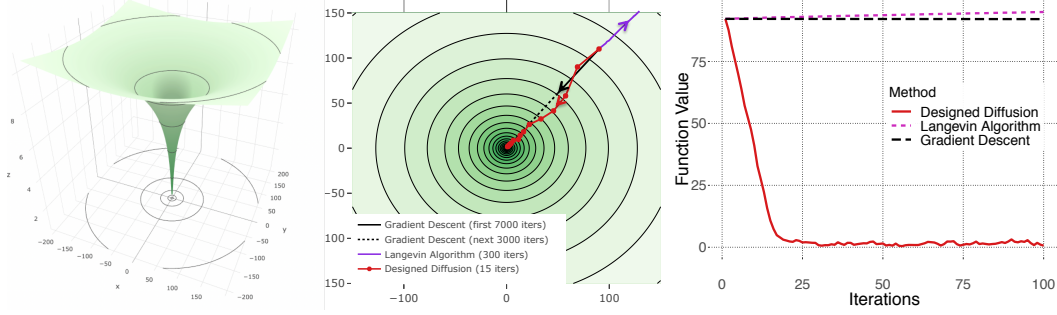

Figure 1: The left plot shows the landscape of the non-convex, sublinear growth function $f(x) = c\log(1 + \frac{1}{2}\|x\|_2^2)$. The middle and right plots compare the optimization error of gradient descent, the Langevin algorithm, and the discretized diffusion designed in Section 5.1.

## 5.1 A simple example with sublinear growth

We begin with a pedagogical example of selecting an appropriate diffusion and verifying our global optimization conditions. Fix $c > \frac{d+3}{2}$ and consider $f(x) = c\log(1 + \frac{1}{2}\|x\|_2^2)$, a simple non-convex objective which exhibits sublinear growth in $\|x\|_2$ and hence does not satisfy dissipativity (Condition 2) when paired with the Gibbs measure Langevin diffusion ($b = -\nabla f, \sigma = \sqrt{2/\gamma}I$). To target the Gibbs measure (2.3) with inverse temperature $\gamma \geq 1$, we choose the diffusion with coefficients $b_\gamma(x) = -\frac{1}{2}a(x)\nabla f(x) + \frac{1}{2\gamma}\langle \nabla, a(x)\rangle$ and $\sigma_\gamma(x) = \frac{1}{\sqrt{\gamma}}\sigma(x)$ for $\sigma(x) \triangleq \sqrt{1 + \frac{1}{2}\|x\|_2^2}I$ and $a(x) = \sigma(x)\sigma(x)^\top$. This choice satisfies Condition 1 with $\lambda_b = \mathcal{O}(1)$, $\lambda_\sigma = \mathcal{O}(\gamma^{-1/2})$, and $\lambda_a = \mathcal{O}(\gamma^{-1})$ with respect to $\gamma$ and Condition 2 with $\alpha = c - \frac{d+3}{2\gamma}$ and $\beta = d/\gamma$. In fact, this diffusion satisfies uniform dissipativity,

$$2\langle b_\gamma(x) - b_\gamma(y), x - y\rangle + \|\sigma_\gamma(x) - \sigma_\gamma(y)\|_{\mathrm{F}}^2,$$
$$= -(c - \tfrac{1}{\gamma})\|x - y\|_2^2 + \tfrac{d}{\gamma}\left(\sqrt{1 + \tfrac{1}{2}\|x\|_2^2} - \sqrt{1 + \tfrac{1}{2}\|y\|_2^2}\right)^2 \leq -\alpha\|x - y\|_2^2,$$

yielding $L_1$ and $L_2$-Wasserstein rates $\varrho_1(t) = \varrho_2(t) = e^{-t\alpha/2}$ by Prop. 3.3 and the relative rate $\tilde{\varrho}_2(t) = 0$. Hence, the $i$-th Stein factor in Theorem 3.2 satisfies $\zeta_i = \mathcal{O}(\gamma^{(i-1)/2})$. This implies that the coefficients $c_i$ in Corollary 4.2 scale with $\mathcal{O}\left(\frac{1}{M\eta} + \Sigma_{i=1}^3 \eta^i \gamma^{i/2} + \frac{1}{\gamma}\right)$ and the final optimization error bound (4.3) can be made of order $\epsilon$ by choosing the inverse temperature $\gamma = \mathcal{O}(\epsilon^{-1})$, the step size $\eta = \mathcal{O}(\epsilon^{1.5})$, and the number of iterations $M = \mathcal{O}(\epsilon^{-2.5})$.

Figure 1 illustrates the value of this designed diffusion over gradient descent and the standard Langevin algorithm. Here, $d = 2$, $c = 10$, the inverse temperature $\gamma = 1$, the step size $\eta = 0.1$, and each algorithm is run from the initial point $(90, 110)$. We observe that the Langevin algorithm diverges, and gradient descent requires thousands of iterations to converge while the designed diffusion converges to the region of interest after 15 iterations.

## 5.2 Non-convex learning with linear growth

Next consider the canonical learning problem of regularized loss minimization with

$$f(x) = \mathcal{L}(x) + \mathcal{R}(x)$$

for $\mathcal{L}(x) \triangleq \frac{1}{L}\sum_{l=1}^L \psi_l(\langle x, v_l\rangle)$, $\psi_l$ a datapoint-specific loss function, $v_l \in \mathbb{R}^d$ the $l$-th datapoint covariate vector, and $\mathcal{R}(x) = \rho(\frac{1}{2}\|x\|_2^2)$ a regularizer with concave $\rho$ satisfying $\delta_3\rho'(z) \geq \sqrt{\max(0, -\rho'(0)z\rho'''(z))}$ and $\frac{4g_1'(z)^2}{\delta_2} \leq \frac{g_1(z)}{z} \leq \delta_1$ for $g_s(z) \triangleq \frac{\rho'(0)}{\rho'(\frac{1}{2}z)} - s$, some $\delta_1, \delta_2, \delta_3 > 0$, and all $z, s \in \mathbb{R}$. Our aim is to select diffusion and stream coefficients that satisfy the Wasserstein decay preconditions of Prop. 3.5. To achieve this, we set $c \equiv 0$ and choose $\sigma$ with $\mu_0(\sigma^{-1}) < \infty$ so

that the regularization component of the drift is *one-sided Lipschitz*, i.e.,

$$-\langle a(x)\nabla\mathcal{R}(x) - a(y)\nabla\mathcal{R}(y), x - y\rangle \leq -K_a\|x-y\|_2^2 \quad \text{for some} \quad K_a > 0. \qquad (5.1)$$

We then show that $L^*$ from Prop. 3.5 is bounded and that, for suitable loss choices, $a(x)\nabla\mathcal{L}(x)$ is bounded and Lipschitz so that (3.5) holds with $K_m = \frac{K_a}{2}$ and $L_m, R_m$ sufficiently large.

Fix any $x$, let $r = \|x\|_2$, and define $\tilde{\sigma}^{(s)}(x) = \sqrt{1-s}(I - \frac{xx^\top}{r^2}) + \frac{xx^\top}{r^2}\sqrt{g_s(r^2)}$ for all $s \in [0,1]$. We choose $\sigma = \tilde{\sigma}^{(0)}$ so that $a(x)\nabla\mathcal{R}(x) = \rho'(0)x$ and (5.1) holds with $K_a = \rho'(0)$. Our constraints on $\rho$ ensure that $a(x) = I + \frac{xx^\top}{r^2}g_1(r^2)$ is positive definite, that $\mu_0(\sigma^{-1}) \leq 1$, and that $\sigma$ and $a$ have at most linear and quadratic growth respectively, in satisfaction of Condition 1. Moreover,

$$\nabla\langle\nabla, a(x)\rangle = I(\tfrac{(d-1)g_1(r^2)}{r^2} + 2g_1'(r^2)) + 2\tfrac{xx^\top}{r^2}((d-1)(g_1'(r^2) - \tfrac{g_1(r^2)}{r^2}) + 2r^2g_1''(r^2)), \text{ and}$$

$$\lambda_{\max}(\nabla\langle\nabla, a(x)\rangle) = \max(\tfrac{(d-1)g_1(r^2)}{r^2} + 2g_1'(r^2), -\tfrac{(d-1)g_1(r^2)}{r^2} + 2dg_1'(r^2) + 4r^2g_1''(r^2)),$$

so that $\lambda_{\max}(\nabla\langle\nabla, a(x)\rangle) \leq \max((d-1)\delta_1 + \sqrt{\delta_1\delta_2}, d\sqrt{\delta_1\delta_2} + 2\delta_3)$. For any $s_0 \in (0,1)$, we have

$$\nabla\tilde{\sigma}^{(s_0)}(x)[v] = (I\tfrac{\langle x,v\rangle}{r} + \tfrac{xv^\top}{r} - 2\tfrac{xx^\top}{r^3}\langle x,v\rangle)\tfrac{\sqrt{g_{s_0}(r^2)}-\sqrt{1-s_0}}{r} + 2\tfrac{xx^\top}{r^3}\langle x,v\rangle\tfrac{rg_{s_0}'(r^2)}{\sqrt{g_{s_0}(r^2)}}$$

for each $v \in \mathbb{R}^d$, so, as $|\sqrt{g_{s_0}(r^2)} - \sqrt{1-s_0}| \leq \sqrt{g_1(r^2)}$, $\phi_1(\tilde{\sigma}) \leq d\sqrt{\delta_1} + \sqrt{\delta_2}$ for $\tilde{\sigma} = \tilde{\sigma}^{(s_0)}$.

Finally, to satisfy (3.5), it suffices to verify that $a(x)\nabla\mathcal{L}(x)$ is bounded and Lipschitz. For example, in the case of a ridge regularizer, $\mathcal{R}(x) = \frac{\lambda}{2}\|x\|_2^2$ for $\lambda > 0$, the coefficient $a(x) = I$, and it suffices to check that $\mathcal{L}$ is Lipschitz with Lipschitz gradient. This strongly convex regularizer satisfies our assumptions, but strong convexity is by no means necessary. Consider instead the pseudo-Huber function, $\mathcal{R}(x) = \lambda(\sqrt{1 + \frac{1}{2}\|x\|_2^2} - 1)$, popularized in computer vision [17]. This convex but non-strongly convex regularizer satisfies all of our criteria and yields a diffusion with $a(x) = I + \frac{xx^\top}{r^2}\frac{\mathcal{R}(x)}{\lambda}$. Moreover, since $\nabla\mathcal{L}(x) = \frac{1}{L}\sum_l v_l\psi_l'(\langle x, v_l\rangle)$ and $\nabla^2\mathcal{L}(x) = \frac{1}{L}\sum_l v_l v_l^\top \psi_l''(\langle x, v_l\rangle)$, $a(x)\nabla\mathcal{L}(x)$ is bounded and Lipschitz whenever $|\psi_l'(r)| \leq \frac{\delta_4}{1+r}$ and $|\psi_l''(r)| \leq \frac{\delta_5}{1+r}$ for some $\delta_4, \delta_5 > 0$. Hence, Prop. 3.5 guarantees exponential Wasserstein decay for a variety of non-convex $\mathcal{L}$ based on datapoint outcomes $y_l$, including the sigmoid ($\psi(r) = \tanh((r-y_l)^2)$ for $y_l \in \mathbb{R}$ or $\psi(r) = 1 - \tanh(y_l r)$ for $y_l \in \{\pm 1\}$) [1], the Student's t negative log likelihood ($\psi_l(r) = \log(1 + (r-y_l)^2)$), and the Blake-Zisserman ($\psi(r) = -\log(e^{-(r-y_l)^2} + \epsilon), \epsilon > 0$) [17]. The reader can verify that all of these examples also satisfy the remaining global optimization pre-conditions of Corollary 4.2 and Theorem 3.2. In contrast, these linear-growth examples do not satisfy dissipativity (Condition 2) when paired with the Gibbs measure Langevin diffusion.

## 6 Conclusion

In this paper, we showed that the Euler discretization of any smooth and dissipative diffusion can be used for global non-convex optimization. We established non-asymptotic bounds on global optimization error and integration error with convergence governed by Stein factors obtained from the solution of the Poisson equation. We further provided explicit bounds on Stein factors for large classes of convex and non-convex objective functions, based on computable properties of the objective and the diffusion. Using this flexibility, we designed suitable diffusions for optimizing non-convex functions not covered by the existing Langevin theory. We also demonstrated that targeting distributions other than the Gibbs measure can give rise to improved optimization guarantees.

## Footnotes

[1]In a typical example where $f$ is bounded by a quadratic polynomial, we have $n = 1$ and $n_e = 6$. We also remind the reader that the double factorial $(n_e - 1)!! = 1 \cdot 3 \cdot 5 \cdots (n_e - 1)$ is of order $\sqrt{n_e!}$.

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
