[Supplementary Material]

# A  Proof of Theorem 3.1: Integration error of discretized diffusions

*Proof of Theorem 3.1.* Denoting by $\Delta X_m = X_{m+1} - X_m$ and using the integral form Taylor's theorem on $u_f(X_{m+1})$ around the previous iterate $X_m$, and taking expectations, we obtain

$$\mathbb{E}[u_f(X_{m+1}) - u_f(X_m)] = \mathbb{E}[\langle \nabla u_f(X_m), \Delta X_m\rangle] + \tfrac{1}{2}\mathbb{E}[\langle \Delta X_m, \nabla^2 u_f(X_m)\Delta X_m\rangle] \tag{A.1}$$
$$+ \tfrac{1}{6}\mathbb{E}[\langle \Delta X_m, \nabla^3 u_f(X_m)[\Delta X_m, \Delta X_m]\rangle]$$
$$+ \tfrac{1}{6}\int_0^1 (1-\tau)^3 \mathbb{E}[\langle \Delta X_m, \nabla^4 u_f(X_m + \tau\Delta X_m)[\Delta X_m, \Delta X_m, \Delta X_m]\rangle]d\tau.$$

The first term on the right hand side can be written as

$$\mathbb{E}[\langle \nabla u_f(X_m), \Delta X_m\rangle] = \mathbb{E}[\langle \nabla u_f(X_m), \eta b(X_m) + \sqrt{\eta}\sigma(X_m)Z_m\rangle],$$
$$= \eta\mathbb{E}[\langle \nabla u_f(X_m), b(X_m)\rangle] + \sqrt{\eta}\,\mathbb{E}[\langle \nabla u_f(X_m), \sigma(X_m)Z_m\rangle],$$
$$= \eta\mathbb{E}[\langle \nabla u_f(X_m), b(X_m)\rangle],$$

where in the last step, we used the fact that $Z_m$ is independent from $X_m$ and that odd moments of $Z_m$ are 0. Similarly for the second and the third terms, we obtain respectively

$$\tfrac{1}{2}\mathbb{E}[\langle \Delta X_m, \nabla^2 u_f(X_m)\Delta X_m\rangle]$$
$$= \tfrac{\eta^2}{2}\mathbb{E}[\langle b(X_m), \nabla^2 u_f(X_m)b(X_m)\rangle] + \tfrac{\eta}{2}\mathbb{E}[\langle \nabla^2 u_f(X_m), \sigma\sigma^\top(X_m)\rangle],$$

and

$$\tfrac{1}{6}\mathbb{E}[\langle \Delta X_m, \nabla^3 u_f(X_m)[\Delta X_m]\Delta X_m\rangle]$$
$$= \tfrac{\eta^3}{6}\mathbb{E}[\langle b(X_m), \nabla^3 u_f(X_m)[b(X_m)]b(X_m)\rangle] + \tfrac{\eta^2}{2}\mathbb{E}[\langle \nabla^3 u_f(X_m)[b(X_m)], \sigma\sigma^\top(X_m)\rangle].$$

By combining these with (3.2), we find that (A.1) can be written as

$$\mathbb{E}[u_f(X_{m+1}) - u_f(X_m)] = \eta\{\mathbb{E}[f(X_m)] - p(f)\} + \tfrac{\eta^2}{2}\mathbb{E}[\langle b(X_m), \nabla^2 u_f(X_m)b(X_m)\rangle]$$
$$+ \tfrac{\eta^3}{6}\mathbb{E}[\langle b(X_m), \nabla^3 u_f(X_m)[b(X_m)]b(X_m)\rangle]$$
$$+ \tfrac{\eta^2}{2}\mathbb{E}[\langle \nabla^3 u_f(X_m)[b(X_m)], \sigma\sigma^\top(X_m)\rangle]$$
$$+ \tfrac{1}{6}\int_0^1 (1-\tau)^3 \mathbb{E}[\langle \Delta X_m, \nabla^4 u_f(X_m + \tau\Delta X_m)[\Delta X_m, \Delta X_m]\Delta X_m\rangle]d\tau.$$

Finally, dividing each term by $\eta$, averaging over $m$, and using the triangle inequality, we reach the bound

$$\left|\tfrac{1}{M}\sum_{m=1}^M \mathbb{E}[f(X_m)] - p(f)\right| \tag{A.2}$$
$$\leq \tfrac{1}{M\eta}\left|\sum_{m=1}^M \mathbb{E}[u_f(X_{m+1}) - u_f(X_m)]\right| + \tfrac{\eta}{2M}\left|\sum_{m=1}^M \mathbb{E}[\langle b(X_m), \nabla^2 u_f(X_m)b(X_m)\rangle]\right|$$
$$+ \tfrac{\eta^2}{6M}\left|\sum_{m=1}^M \mathbb{E}[\langle b(X_m), \nabla^3 u_f(X_m)[b(X_m)]b(X_m)\rangle]\right|$$
$$+ \tfrac{\eta}{2M}\left|\sum_{m=1}^M \mathbb{E}[\langle \nabla^3 u_f(X_m)[b(X_m)], \sigma\sigma^\top(X_m)\rangle]\right|$$
$$+ \tfrac{1}{6M\eta}\left|\sum_{m=1}^M \int_0^1 (1-\tau)^3 \mathbb{E}[\langle \Delta X_m, \nabla^4 u_f(X_m + \tau\Delta X_m)[\Delta X_m, \Delta X_m]\Delta X_m\rangle]d\tau\right|.$$

For the first term on the right hand side, using Condition 3 and Lemma A.2, we can write

$$\left|\sum_{m=1}^M \mathbb{E}[u_f(X_{m+1}) - u_f(X_m)]\right| = |\mathbb{E}[u_f(X_{M+1}) - u_f(X_1)]| \tag{A.3}$$
$$\leq \tilde{\mu}_{1,n}(u_f)\mathbb{E}[(1 + \|X_{M+1}\|_2^n + \|X_1\|_2^n)\|X_{M+1} - X_1\|_2],$$
$$\leq \tilde{\mu}_{1,n}(u_f)\mathbb{E}[2 + 3\|X_{M+1}\|_2^{n+1} + 3\|X_1\|_2^{n+1}],$$
$$\leq 6\tilde{\mu}_{1,n}(u_f)\left(2 + \tfrac{2\beta_{r,n_e}}{\alpha} + \|x\|_2^{n_e}\right).$$

where we used Young's inequality in the second step and Lemma A.2 in the last step.

The second term in the above inequality can be bounded by

$$\frac{\eta}{2M}\left|\sum_{m=1}^{M}\mathbb{E}\big[\langle b(X_m),\nabla^2 u_f(X_m)b(X_m)\rangle\big]\right| \le \frac{\eta}{2M}\sum_{m=1}^{M}\mathbb{E}\big[|\langle b(X_m),\nabla^2 u_f(X_m)b(X_m)\rangle|\big] \tag{A.4}$$

$$\le \frac{\eta}{2M}\sum_{m=1}^{M}\mathbb{E}\big[\|\nabla^2 u_f(X_m)\|_{\mathrm{op}}\|b(X_m)\|_2^2\big]$$

$$\le \frac{\eta\lambda_b^2}{32M}\sum_{m=1}^{M}\mathbb{E}\big[\zeta_2(1+\|X_m\|_2^n)(1+\|X_m\|_2)^2\big]$$

$$\le \frac{\eta\lambda_b^2\zeta_2}{8M}\sum_{m=1}^{M}\mathbb{E}\big[1+\|X_m\|_2^{n+2}\big]$$

$$\le \frac{\eta\lambda_b^2\zeta_2}{8}\Big(2+\frac{2\beta_{r,n_e}}{\alpha}+\|x\|_2^{n_e}\Big),$$

where in the last step we used Lemma A.2.

Similarly, the third and the fourth terms in the inequality (A.2) can be bounded as

$$\frac{\eta^2}{6M}\left|\sum_{m=1}^{M}\mathbb{E}\big[\langle b(X_m),\nabla^3 u_f(X_m)[b(X_m)]b(X_m)\rangle\big]\right| \le \frac{\eta^2\lambda_b^3\zeta_3}{48M}\sum_{m=1}^{M}\mathbb{E}\big[1+\|X_m\|_2^{n+3}\big]$$

$$\le \frac{\eta^2\lambda_b^3\zeta_3}{48}\Big(2+\frac{2\beta_{r,n_e}}{\alpha}+\|x\|_2^{n_e}\Big), \tag{A.5}$$

and

$$\frac{\eta}{2M}\left|\sum_{m=1}^{M}\mathbb{E}\big[\langle\nabla^3 u_f(X_m)[b(X_m)],\sigma\sigma^\top(X_m)\rangle\big]\right| \le \frac{\eta\lambda_b\lambda_\sigma^2\zeta_3}{16M}\sum_{m=1}^{M}\mathbb{E}\big[1+\|X_m\|_2^{n+3}\big]$$

$$\le \frac{\eta\lambda_b\lambda_\sigma^2\zeta_3}{16}\Big(2+\frac{2\beta_{r,n_e}}{\alpha}+\|x\|_2^{n_e}\Big). \tag{A.6}$$

For the last term, we write

$$\frac{1}{6\eta M}\left|\sum_{m=1}^{M}\mathbb{E}\Big[\int_0^1(1-\tau)^3\langle\Delta X_m,\nabla^4 u_f(X_m+\tau\Delta X_m)[\Delta X_m,\Delta X_m]\Delta X_m\rangle d\tau\Big]\right|$$

$$\le \frac{1}{6\eta M}\sum_{m=1}^{M}\int_0^1(1-\tau)^3\zeta_4\mathbb{E}\big[(1+\|X_m+\tau\Delta X_m\|_2^n)\|\Delta X_m\|_2^4\big]d\tau.$$

We first bound the expectation in the above integral

$$\mathbb{E}\big[(1+\|X_m+\tau\Delta X_m\|_2^n)\|\Delta X_m\|_2^4\big] \le \mathbb{E}\Big[8(\eta^4\|b(X_m)\|_2^4+\eta^2\|\sigma(X_m)W_m\|_2^4) \tag{A.7}$$

$$\times \big(1+3^{n-1}\|X_m\|_2^n+3^{n-1}\tau^n\eta^n\|b(X_m)\|_2^n+3^{n-1}\tau^n\eta^{n/2}\|\sigma(X_m)W_m\|_2^n\big)\Big]$$

$$= A+\tau^n B, \quad\text{where}$$

$$A\triangleq 8\mathbb{E}\big[(1+3^{n-1}\|X_m\|_2^n)(\eta^4\|b(X_m)\|_2^4+\eta^2\|\sigma(X_m)W_m\|_2^4)\big]$$

$$B\triangleq 8\,3^{n-1}\mathbb{E}\big[(\eta^n\|b(X_m)\|_2^n+\eta^{n/2}\|\sigma(X_m)W_m\|_2^n)(\eta^4\|b(X_m)\|_2^4+\eta^2\|\sigma(X_m)W_m\|_2^4)\big].$$

Using Condition 1, Lemma E.1 and $\eta<1$, we obtain

$$A\le 8\Big\{\eta^4\Big[\tfrac{\lambda_b^4}{32}\mathbb{E}\big[1+\|X_m\|_2^4\big]+3^{n-1}\tfrac{\lambda_b^4}{16}\mathbb{E}\big[1+\|X_m\|_2^{n+4}\big]\Big]$$

$$+\eta^2\Big[\tfrac{3\lambda_\sigma^4}{32}\mathbb{E}\big[1+\|X_m\|_2^4\big]+3^n\tfrac{\lambda_\sigma^4}{16}\mathbb{E}\big[1+\|X_m\|_2^{n+4}\big]\Big]\Big\}$$

$$\le \tfrac{1+3^{n-1}}{2}(\eta^4\lambda_b^4+3\eta^2\lambda_\sigma^4)\mathbb{E}\big[1+\|X_m\|_2^{n+4}\big], \quad\text{and}$$

$$B\le 8\,3^{n-1}\eta^{2+n/2}\mathbb{E}\Big[\eta^{2+n/2}\|b(X_m)\|_2^{n+4}+\eta^2\|b(X_m)\|_2^4\|\sigma(X_m)W_m\|_2^n$$

$$+3\|b(X_m)\|_2^n\|\sigma(X_m)\|_{\mathrm{F}}^4+\|\sigma(X_m)W_m\|_2^{n+4}\Big]$$

$$\le \eta^{2+n/2}\tfrac{3^{n-1}}{2^{n+2}}\big(\eta^2\lambda_b^4+(n+4)(n+2)\lambda_\sigma^4\big)\big(\eta^{n/2}\lambda_b^n+n!!\lambda_\sigma^n\big)\mathbb{E}\big[1+\|X_m\|_2^{n+4}\big],$$

$$\le \eta^{2+n/2}\tfrac{1}{12}1.5^n\big(\lambda_b^4+n_e^2\lambda_\sigma^4\big)(\lambda_b^n+n!!\lambda_\sigma^n)\mathbb{E}\big[1+\|X_m\|_2^{n+4}\big].$$

Plugging this in (A.7), we obtain

$$\mathbb{E}\big[(1+\|X_m+\tau\Delta X_m\|_2^n)\|\Delta X_m\|_2^4\big]$$

$$\le \mathbb{E}\big[1+\|X_m\|_2^{n+4}\big]\Big[\tfrac{1+3^{n-1}}{2}(\eta^4\lambda_b^4+3\eta^2\lambda_\sigma^4)+\tau^n\eta^{2+n/2}\tfrac{1}{12}1.5^n\big(\lambda_b^4+n_e^2\lambda_\sigma^4\big)(\lambda_b^n+n!!\lambda_\sigma^n)\Big].$$

Therefore, the last term in (A.2) can be bounded by

$$\frac{1}{6\eta M}\left|\sum_{m=1}^{M}\mathbb{E}\left[\int_0^1 (1-\tau)^3\langle \Delta X_m, \nabla^4 u_f(X_m + \tau\Delta X_m)[\Delta X_m, \Delta X_m]\Delta X_m\rangle d\tau\right]\right| \tag{A.8}$$

$$\leq \frac{\zeta_4\eta}{6}\frac{1}{M}\sum_{m=1}^{M}\mathbb{E}\left[1 + \|X_m\|_2^{n+4}\right]$$

$$\times \int_0^1 (1-\tau)^3\left(\frac{1+3^{n-1}}{2}(\eta^4\lambda_b^4 + 3\eta^2\lambda_\sigma^4) + \tau^n\eta^{2+n/2}\frac{1}{12}1.5^n\left(\lambda_b^4 + n_e^2\lambda_\sigma^4\right)(\lambda_b^n + n!!\lambda_\sigma^n)\right)d\tau.$$

Using Lemma A.2 and

$$\int_0^1 (1-\tau)^3\tau^n d\tau \leq \frac{6}{n^4} \quad\text{and}\quad \int_0^1 (1-\tau)^3 d\tau = \frac{1}{4},$$

the right hand side of (A.8) can be bounded by

$$\frac{\zeta_4\eta}{6}\left(\frac{1+3^{n-1}}{8}(\eta^2\lambda_b^4 + 3\lambda_\sigma^4) + \eta^{n/2}\frac{1}{2n^4}1.5^n\left(\lambda_b^4 + n_e^2\lambda_\sigma^4\right)(\lambda_b^n + n!!\lambda_\sigma^n)\right)\left(2 + \frac{2\beta_{r,n_e}}{\alpha} + \|x\|_2^{n_e}\right). \tag{A.9}$$

Combining the above bounds in (A.3), (A.4), (A.5), (A.6) and (A.9) and applying them on (A.2), we reach the final bound

$$\left|\frac{1}{M}\sum_{m=1}^{M}\mathbb{E}[f(X_m)] - p(f)\right| \leq \left(c_1\frac{1}{\eta M} + c_2\eta + c_3\eta^{1+|1\wedge n/2|}\right)\left(\kappa_r + \|x\|_2^{n_e}\right)$$

where

$$c_1 = 6\zeta_1,$$

$$c_2 = \frac{1}{16}\left[\zeta_2 2\lambda_b^2 + \zeta_3\lambda_b\lambda_\sigma^2 + \zeta_4(1 + 3^{n-1})\lambda_\sigma^4\right],$$

$$c_3 = \frac{1}{48}\left[\zeta_3\lambda_b^3 + \zeta_4\lambda_b^4(1 + 3^{n-1}) + \zeta_4 4\frac{1.5^n}{n^4}(\lambda_b^4 + n_e^2\lambda_\sigma^4)(\lambda_b^n + n!!\lambda_\sigma^n)\right], \quad\text{and}$$

$$\kappa_r = 2 + \frac{2\beta}{\alpha} + \frac{n_e\lambda_a}{4\alpha} + \frac{\tilde{\alpha}_r}{\alpha}\left(\frac{n_e\lambda_a + 6r\beta}{2r\tilde{\alpha}_r}\right)^{n_e},$$

for $\tilde{\alpha}_1 = \alpha$ and $\tilde{\alpha}_2 = [\alpha - n_e\lambda_a/4]_+$. $\qquad\square$

## A.1 Dissipativity for higher order moments

It is well known that the dissipativity condition on the second moment carries directly to the higher order moments [22]. The following lemma will be useful when we bound the higher order moments of the discretized diffusion.

**Lemma A.1.** *For $n \geq k \geq 2$, we have the following relation*

$$\mathcal{A}\|x\|_2^n = \frac{n}{k}\|x\|_2^{n-k}\mathcal{A}\|x\|_2^k + \frac{1}{2}n(n-k)\|x\|_2^{n-4}\|\sigma^\top(x)x\|_2^2.$$

*Further, assume that Conditions 1 and 2 hold, and $n \geq 3$. Then,*

$$\mathcal{A}\|x\|_2^n \leq -\alpha\|x\|_2^n + \beta_{r,n}$$

*where*

$$\beta_{r,n} = \beta + \frac{n\lambda_a}{8} + \frac{\tilde{\alpha}_r}{2}\left(\frac{n\lambda_a + 6r\beta}{2r\tilde{\alpha}_r}\right)^n,$$

*with $\tilde{\alpha}_2 = [\alpha - n\lambda_a/4]_+$ and $\tilde{\alpha}_1 = \alpha$.*

*Proof.* The proof for the first statement easily follows from the following expression,

$$\mathcal{A}\|x\|_2^n = n\|x\|_2^{n-2}\langle x, b(x)\rangle + \frac{1}{2}n(n-2)\|x\|_2^{n-4}\langle xx^\top, \sigma\sigma^\top(x)\rangle + \frac{1}{2}n\|x\|_2^{n-2}\|\sigma(x)\|_F^2.$$

For second statement, we use the first statement with $k = 2$ and Conditions 1 and 2. First, we consider the case $r = 1$ and write

$$\mathcal{A}\|x\|_2^n = \frac{1}{2}n\|x\|_2^{n-2}\mathcal{A}\|x\|_2^2 + \frac{1}{2}n(n-2)\|x\|_2^{n-4}\langle xx^\top, \sigma\sigma^\top(x)\rangle,$$

$$\leq -\frac{1}{2}\alpha n\|x\|_2^n + \frac{1}{2}\beta n\|x\|_2^{n-2} + \frac{\lambda_a}{8}n(n-2)(\|x\|_2^{n-1} + \|x\|_2^{n-2}),$$

$$= -\frac{1}{2}\alpha n\|x\|_2^n + \frac{\lambda_a}{8}n(n-2)\|x\|_2^{n-1} + \left\{\frac{1}{2}\beta n + \frac{\lambda_a}{8}n(n-2)\right\}\|x\|_2^{n-2}.$$

Using the inequality given in Lemma E.3 twice, we obtain

$$\mathcal{A}\|x\|_2^n \leq -\frac{1}{2}\alpha n\|x\|_2^n + \left\{\frac{\lambda_a}{4}n(n-2) + \frac{1}{2}\beta n\right\}\|x\|_2^{n-1} + \beta + \frac{\lambda_a n}{8},$$

$$\leq -\alpha\|x\|_2^n + \frac{\alpha(n-2)}{2n}\left(\frac{n\lambda_a}{2\alpha} + \frac{\beta n}{\alpha(n-2)}\right)^n + \frac{n(n-2)\lambda_a}{8(n-1)} + \frac{n\beta}{2(n-1)}.$$

Same calculation yields a similar expression for the case $r = 2$. Generalizing, we obtain the following formula,

$$\mathcal{A}\|x\|_2^n \leq -\alpha\|x\|_2^n + \frac{\alpha_r(n-2)}{2n}\left(\frac{n\lambda_a}{2r\alpha_r} + \frac{n\beta}{(n-2)\alpha_r}\right)^n + \frac{n(n-2)\lambda_a}{8(n-1)} + \frac{n\beta}{2(n-1)},$$

$$\leq -\alpha\|x\|_2^n + \frac{\alpha_r}{2}\left(\frac{n\lambda_a + 6r\beta}{2r\alpha_r}\right)^n + \frac{n\lambda_a}{8} + \beta.$$

$\qquad\square$

## A.2 Proof of Lemma A.2: Markov Chain Moment Bounds

**Lemma A.2.** *Let the Conditions 1 and 2 hold. For $n \geq 1$, denote by $n_e$ an even integer satisfying $n_e \geq n$. If the step size satisfies*

$$\eta < 1 \wedge \frac{\alpha}{2(n_e-1)!!(1+\lambda_b/2+\lambda_\sigma/2)^{n_e}},$$

*then we have*

$$\mathbb{E}[\|X_m\|_2^n] \leq \|x\|_2^{n_e} + 1 + \frac{2\beta_{r,n_e}}{\alpha},$$
$$\frac{1}{M}\sum_{m=1}^{M}\mathbb{E}[\|X_m\|_2^n] \leq \|x\|_2^{n_e} + 1 + \frac{2\beta_{r,n_e}}{\alpha}.$$

*Proof of Lemma A.2.* First, we handle the even moments. For $n \geq 1$, we write

$$\mathbb{E}\big[\|X_m + \eta b(X_m) + \sqrt{\eta}\sigma(X_m)W_m\|_2^{2n}\big] = \mathbb{E}\Big[\big(\|X_m\|_2^2 + \eta^2\|b(X_m)\|_2^2 + \eta\|\sigma(X_m)W_m\|_2^2$$
$$+ 2\eta\langle X_m, b(X_m)\rangle + 2\eta^{0.5}\langle X_m, \sigma(X_m)W_m\rangle + 2\eta^{1.5}\langle b(X_m),\sigma(X_m)W_m\rangle\big)^n\Big]$$

$$\overset{1}{=} \sum_{k_1+k_2+\ldots+k_6=n}\binom{n}{k_1,k_2,\ldots,k_6}\eta^{2k_2+k_3+k_4+k_5/2+3k_6/2}\,2^{k_4+k_5+k_6}$$
$$\mathbb{E}\Big[\|X_m\|_2^{2k_1}\|b(X_m)\|_2^{2k_2}\|\sigma(X_m)W_m\|_2^{2k_3}\langle X_m,b(X_m)\rangle^{k_4}\langle X_m,\sigma(X_m)W_m\rangle^{k_5}\langle b(X_m),\sigma(X_m)W_m\rangle^{k_6}\Big],$$

$$\overset{2}{\leq}\mathbb{E}\big[\|X_m\|_2^{2n}\big]+\eta\mathbb{E}\big[\mathcal{A}\|X_m\|_2^{2n}\big]+\sum_{\substack{k_1+k_2+\ldots+k_6=n\\ k_5+k_6 \text{ is even}\\ 2k_2+k_3+k_4+k_5/2+3k_6/2>1}}\binom{n}{k_1,k_2,\ldots,k_6}\eta^{2k_2+k_3+k_4+k_5/2+3k_6/2}\,2^{k_4+k_5+k_6}$$
$$\mathbb{E}\Big[\|X_m\|_2^{2k_1+k_4+k_5}\|b(X_m)\|_2^{2k_2+k_4+k_6}\|\sigma(X_m)W_m\|_2^{2k_3+k_5+k_6}\Big]$$

$$\overset{3}{\leq}\sum_{\substack{k_1+k_2+\ldots+k_6=n\\ k_5+k_6 \text{ is even}\\ 2k_2+k_3+k_4+k_5/2+3k_6/2>1}}\binom{n}{k_1,k_2,\ldots,k_6}(2k_3+k_5+k_6-1)!!\eta^{2k_2+k_3+k_4+k_5/2+3k_6/2}\,2^{k_4+k_5+k_6}$$
$$\mathbb{E}\Big[\|X_m\|_2^{2k_1+k_4+k_5}\|b(X_m)\|_2^{2k_2+k_4+k_6}\|\sigma(X_m)\|_F^{2k_3+k_5+k_6}\Big]+(1-\eta\alpha)\mathbb{E}\big[\|X_m\|_2^{2n}\big]+\eta\beta_{r,2n}$$

$$\overset{4}{\leq}(1-\eta\alpha+\eta^2 2\rho_n)\mathbb{E}\big[\|X_m\|_2^{2n}\big]+\eta\beta_{r,2n}+\eta^2\rho_n$$

where

$$\rho_n = \frac{1}{2}\sum_{\substack{k_1+k_2+\ldots+k_6=n\\ k_5+k_6 \text{ is even}\\ 2k_2+k_3+k_4+k_5/2+3k_6/2>1}}\binom{n}{k_1,k_2,\ldots,k_6}(2k_3+k_5+k_6-1)!!\frac{\lambda_b^{2k_2+k_4+k_6}\lambda_\sigma^{2k_3+k_5+k_6}}{2^{2k_2+2k_3+k_6}}.$$

In the above derivation, step (1) follows from multinomial expansion theorem, step (2) follows from that the odd moments of a Gaussian random variable is 0, and that the terms with coefficient $\eta$ add up to $\mathbb{E}\big[\mathcal{A}\|X_m\|_2^{2n}\big]$. Step (3) follows from Cauchy-Schwartz, Lemma E.1, and Condition 2, and finally step (4) uses Condition 1 and the fact that $\eta < 1$.

A compact and more interpretable estimate for $\rho_n$ can be obtained as follows,

$$\rho_n \leq \frac{1}{2}(2n-1)!!\sum_{k_1+k_2+\ldots+k_6=n}\binom{n}{k_1,k_2,\ldots,k_6}\frac{\lambda_b^{2k_2+k_4+k_6}\lambda_\sigma^{2k_3+k_5+k_6}}{2^{2k_2+2k_3+k_6}}$$
$$= \frac{1}{2}(2n-1)!!(1+\frac{\lambda_b}{2}+\frac{\lambda_\sigma}{2})^{2n}.$$

The above result reads

$$\mathbb{E}\big[\|X_{m+1}\|_2^{2n}\big] \leq \tau_n(\eta)\mathbb{E}\big[\|X_m\|_2^{2n}\big]+\tilde{\gamma}_n(\eta)$$

where $\tau_n(\eta) = 1 - \eta\alpha + \eta^2 2\rho_n$, and $\tilde{\gamma}_n(\eta) = \eta\beta_{r,2n}+\eta^2\rho_n$. Notice that $\tau_n(0)=1$ and $\tau_n'(0)=-\alpha$ is negative. Therefore, we may obtain $\tau_n(\eta) < 1$ by choosing $\eta$ small. More specifically, we have $\tau_n(\eta) < 1$ when $\eta < \alpha/2\rho_n$, but by choosing $\eta < \alpha/(4\rho_n)$ we have control over the second term as well. That is, by Lemma E.2, we immediately obtain

$$\mathbb{E}\big[\|X_m\|_2^{2n}\big] \leq \tau_n(\eta)^m\|x\|^{2n}+\frac{\tilde{\gamma}_n(\eta)}{1-\tau_{2n}(\eta)}$$
$$\leq \tau_n(\eta)^m\|x\|^{2n}+\frac{2\beta_{r,2n}+\alpha/2}{\alpha}$$
$$\leq \|x\|^{2n}+\frac{2\beta_{r,2n}+\alpha}{\alpha}$$

and

$$\tfrac{1}{M}\sum_{m=1}^{M}\mathbb{E}\big[\|X_m\|_2^{2n}\big] \le \|x\|^{2n} + \tfrac{2\beta_{r,2n}+\alpha}{\alpha}$$

where we use a looser bound to ensure that the right hand side is larger than 1.

The above analysis only covers the even moments so far. For any integer $n$, denote by $n_e$ an even integer that is not smaller than $n$. Then, by the Hölder's inequality we write

$$\mathbb{E}[\|X_m\|_2^{n}] \le \mathbb{E}[\|X_m\|_2^{n_e}]^{n/n_e} \le \Big( \|x\|^{n_e} + \tfrac{2\beta_{r,n_e}+\alpha}{\alpha} \Big)^{n/n_e}$$

$$\le \|x\|^{n_e} + \tfrac{2\beta_{r,n_e}}{\alpha} + 1,$$

which concludes the proof.

$\square$

## B  Proof of Theorem 3.2: Stein Factor Bounds

Let $(P_t)_{t=0}^{\infty}$ denote the transition semigroup of the diffusion $(Z_t^x)_{t=0}^{\infty}$ with drift and diffusion coeffieients $b,\sigma$, so that $(P_t f)(x) = \mathbb{E}[f(Z_t^x)]$ for each $x \in \mathbb{R}^d$ and $t \ge 0$. Define the function

$$\varphi_{i,n}(b,\sigma) \triangleq \mu_i(b) + n\mu_i(\sigma)^2 + \phi_i(\sigma)^2$$

and the constants

$$\begin{aligned}
\gamma_{1,n} &= 1,\\
\theta_{1,n} &= n\varphi_{1,n-2}(b,\sigma),\\
\gamma_{2,n} &= \tfrac{\varphi_{2,n-2}(b,\sigma)}{n\varphi_{1,2n-2}(b,\sigma)}\\
\theta_{2,n} &= 3n\varphi_{1,2n-2}(b,\sigma) + n\varphi_{2,n-2}(b,\sigma),\\
\gamma_{3,n} &= \tfrac{15\varphi_{2,n-2}(b,\sigma)+5\varphi_{3,n-2}(b,\sigma)}{4n\varphi_{1,4n-2}(b,\sigma)},\\
\theta_{3,n} &= 7n\varphi_{1,3n-2}(b,\sigma) + 10n\varphi_{2,n-2}(b,\sigma) + 3n\varphi_{3,n-2}(b,\sigma),\\
\gamma_{4,2} &= \tfrac{\varphi_{4,0}(b,\sigma)+6\varphi_{3,0}(b,\sigma)+5\varphi_{2,0}(b,\sigma)}{16\varphi_{1,6}(b,\sigma)}, \quad \text{and}\\
\theta_{4,2} &= 31\varphi_{1,5}(b,\sigma) + 27\varphi_{2,2}(b,\sigma) + 12\varphi_{3,1}(b,\sigma) + \varphi_{4,0}(b,\sigma).
\end{aligned}$$

Our proof of Theorem 3.2 will use a representation of the Poisson equation solution in terms of the transition semigroup, i.e.,

$$u_f(x) = \int_0^{\infty} p(f) - (P_t f)(x)dt, \tag{B.1}$$

coupled with the following bounds on the derivatives of the semigroup. See [12] for a proof of the above representation.

**Theorem B.1** (Semigroup derivative bounds [12])**.** *Let* $(P_t)_{t=0}^{\infty}$ *denote the transition semigroup of a diffusion with drift and diffusion coeffieients $b$ and $\sigma$. Define* $\tilde{\varrho}_1(t) = \log(\varrho_2(t)/\varrho_1(t))$, $\tilde{\varrho}_2(t) = [\log(\varrho_1(t)/\varrho_2(t)/\varrho_1(0))]/\log(\varrho_1(t)/\varrho_1(0))$, $\tilde{\alpha}_1 = \alpha$, *and* $\tilde{\alpha}_2 = \inf_{t\ge 0}[\alpha - n\lambda_a(1 \vee \tilde{\varrho}_2(t))]_+$. *If $f : \mathbb{R}^d \to \mathbb{R}$ is pseudo-Lipschitz continuous of order $n$ then $P_t f$ satisfies the pseudo-Lipschitz bounds*

$$\tilde{\mu}_{1,n}(P_t f) \le \tilde{\mu}_{1,n}(f)\varrho_1(t)\omega_r(t) \quad \text{and} \quad \tilde{\pi}_{1,n}(P_t f) \le 2\tilde{\mu}_{1,n}(f)\varrho_1(t)\omega_r(t) \tag{B.2}$$

*for*

$$\omega_r(t) = 1 + 4\varrho_1(t)^{1-1/r}\varrho_1(0)^{1/2}\Big(1 + 2\Big[\tfrac{[1\vee\tilde{\varrho}_r(t)]2\lambda_a n + 3r\beta}{\tilde{\alpha}_r}\Big]^n\Big).$$

*Furthermore,* $\nabla^2 P_t f$ *satisfies the degree-$n$ polynomial growth bound*

$$\tilde{\pi}_{2,n}(P_t f) \le \xi_2 \varrho_1(t-1)\omega_r(t-1) \quad \text{for} \tag{B.3}$$

$$\xi_2 = 4\tilde{\mu}_{1,n}(f)\{1 + (\beta_{r,6n}/\alpha)^{1/6}\}\varrho_1(0)\omega_r(1)\tilde{\pi}_{0,0}(\sigma^{-1})\Big[1 + \gamma_{2,2}^{1/2} + \mu_1(\sigma)\Big]e^{\theta_{2,2}/2}.$$

*If, in addition,* $\tilde{\pi}_{2:3,n}(f) < \infty$, $\nabla^3 P_t f$ *satisfies the degree-$n$ polynomial growth bounds, for $t \ge 2$,*

$$\tilde{\pi}_{3,n}(P_t f) \le \xi_3 \varrho_1(t-2)\omega_r(t-2) \quad \text{where} \tag{B.4}$$

$$\xi_3 = 4\tilde{\mu}_{1,n}(f)\tilde{\pi}_{1,0}(\sigma)\tilde{\pi}_{1:2,0}(\sigma)\tilde{\pi}_{0,0}(\sigma^{-1})\tilde{\pi}_{0:1,0}(\sigma^{-1})\varrho_1(0)\omega_r(1)e^{\theta_{3,4}/2}$$

$$\times (7 + 7\gamma_{2,2}^{1/2} + \gamma_{2,3}^{1/3} + \gamma_{3,2}^{1/2})\{1 + (\beta_{r,6n}/\alpha)^{1/6}\}^2,$$

*and, for $t < 2$,*

$$\tilde{\pi}_{3,n}(P_t f) \leq 2\tilde{\pi}_{1:3,n}(f)\big(1 + 3\gamma_{2,3}^{1/3} + \gamma_{3,2}^{1/2}\big)e^{t\theta_{3,4}/4}\big\{1 + (\beta_{r,6n}/\alpha)^{1/6}\big\}. \tag{B.5}$$

*If, in addition, $\tilde{\pi}_{4,n}(f) < \infty$, $\nabla^4 P_t f$ satisfies the degree-$n$ polynomial growth bounds, for $t \geq 3$*

$$\tilde{\pi}_{4,n}(P_t f) \leq \xi_4 \varrho_1(t-3)\omega_r(t-1), \quad \text{where} \tag{B.6}$$

$$\xi_4 = 4\tilde{\mu}_{1,n}(f)\tilde{\pi}_{1,0}(\sigma)^2 \tilde{\pi}_{1:3,0}(\sigma)\tilde{\pi}_{0,0}(\sigma^{-1})^2 \tilde{\pi}_{0:2,0}(\sigma^{-1})e^{\theta_{4,2}}\varrho_1(0)\omega_r(1)\big\{1 + (\beta_{r,6n}/\alpha)^{1/6}\big\}^3$$

$$\times \Big[42 + 32\gamma_{2,2}^{1/2} + 6\gamma_{2,2} + 2\gamma_{2,3}^{1/3} + 3\gamma_{2,3}^{2/3} + 24\gamma_{2,4}^{1/4} + 3\gamma_{2,4}^{1/2} + 12\gamma_{2,6}^{1/6} + 5\gamma_{3,2}^{1/2} + 5\gamma_{3,3}^{1/3} + \gamma_{4,2}^{1/2} + 6\gamma_{2,2}^{1/2}\gamma_{2,6}^{1/6}\Big],$$

*and, for $t < 3$,*

$$\tilde{\pi}_{4,n}(P_t f) \leq 2\tilde{\pi}_{1:4,n}(f)\big\{1 + (\beta_{r,6n}/\alpha)^{1/6}\big\}\Big[1 + 6\gamma_{2,4}^{1/4} + 4\gamma_{2,3} + 3\gamma_{2,3}^{2/3} + 4\gamma_{3,3}^{1/3} + \gamma_{4,2}^{1/2}\Big]e^{t\theta_{4,2}/2}. \tag{B.7}$$

To establish the first Stein factor bound $\zeta_1$, we combine the representation (B.1), the triangle inequality, and the definition of pseudo-Lipschitzness to find that

$$|u_f(x) - u_f(y)| \leq \int_0^\infty |(P_t f)(x) - (P_t f)(y)|dt,$$
$$\leq \int_0^\infty \tilde{\mu}_{1,n}(P_t f)dt(1 + \|x\|_2^n + \|y\|_2^n)\|x - y\|_2.$$

Invoking the pseudo-Lipschitz constant for $P_t f$ (B.2) now yields the first Stein factor bound.

For each additional Stein factor, the dominated convergence theorem will enable us to differentiate under the integral sign. For the second Stein factor $\zeta_2$, using the second derivative of the representation (B.1) and the bound (B.3), we obtain

$$\big|\langle u, \nabla^2 u_f(x)v\rangle\big| \leq \int_0^\infty \big|\langle u, \nabla^2(P_t f)(x)v\rangle\big|dt$$
$$\leq 4\bar{\rho}_2\big\{1 + (\beta_{r,6n}/\alpha)^{1/6}\big\}\tilde{\mu}_{1,n}(f)(1 + \|x\|_2^n)\|u\|_2\|v\|_2,$$

where

$$\bar{\rho}_2 = \int_0^\infty \bar{\xi}_2(1 \wedge t)\varrho_1(t - 1 \wedge t)\omega_r(t - 1 \wedge t)dt$$
$$= \varrho_1(0)\omega_r(0)\int_0^1 \bar{\xi}_2(t)dt + \bar{\xi}_2(1)\int_1^\infty \varrho_1(t-1)\omega_r(t-1)dt,$$
$$= \varrho_1(0)\omega_r(0)\tilde{\pi}_{0,0}(\sigma^{-1})\varrho_1(0)\omega_r(1)e^{\theta_{2,2}/2}\Big[2 + \gamma_{2,2}^{1/2} + \mu_1(\sigma)\Big] + \bar{\xi}_2(1)\int_0^\infty \varrho_1(t)\omega_r(t)dt.$$

The final bound is obtained by taking the supremum over $u$ and $v$, i.e.,

$$\|\nabla^2 u_f(x)\|_{\mathrm{op}} = \sup_{\|u\|_2 = \|v\|_2 = 1} \langle u, \nabla^2 u_f(x)v\rangle \leq \zeta_2(1 + \|x\|_2^n),$$

where

$$\zeta_2 \triangleq 2\xi_2\varrho_1(0)\omega_r(0) + \xi_2\int_0^\infty \varrho_1(t)\omega_r(t)dt, \quad \text{with}$$
$$\xi_2 = 4\tilde{\mu}_{1,n}(f)\big\{1 + (\beta_{r,6n}/\alpha)^{1/6}\big\}\varrho_1(0)\omega_r(1)\tilde{\pi}_{0,0}(\sigma^{-1})\Big[1 + \gamma_{2,2}^{1/2} + \mu_1(\sigma)\Big]e^{\theta_{2,2}/2}.$$

For the third Stein factor $\zeta_3$, using the third derivative of the representation (B.1), and the bounds (B.4) and (B.5), we obtain

$$\big|\nabla^3 u_f(x)[v, u, w]\big| \leq \int_0^2 \big|\nabla^3(P_t f)(x)[v, u, w]\big|dt + \int_2^\infty \big|\nabla^3(P_t f)(x)[v, u, w]\big|dt$$
$$\leq \Big[4\tilde{\pi}_{1:3,n}(f)\big(1 + 3\gamma_{2,3}^{1/3} + \gamma_{3,2}^{1/2}\big)e^{\theta_{3,4}/2}\big\{1 + (\beta_{r,6n}/\alpha)^{1/6}\big\}$$
$$+ \xi_3\int_2^\infty \varrho_1(t-2)\omega_r(t-1)dt\Big](1 + \|x\|_2^n)\|u\|_2\|v\|_2\|w\|_2.$$

Consequently, we obtain

$$\|\nabla^3 u_f(x)\|_{\mathrm{op}} \leq \zeta_3(1 + \|x\|_2^n) \quad \text{where}$$
$$\zeta_3 = 4\tilde{\pi}_{1:3,n}(f)\big(1 + 3\gamma_{2,3}^{1/3} + \gamma_{3,2}^{1/2}\big)\big\{1 + (\beta_{r,6n}/\alpha)^{1/6}\big\} + \xi_3\int_0^\infty \varrho_1(t)\omega_r(t+1)dt \quad \text{and}$$
$$\xi_3 = 4\tilde{\mu}_{1,n}(f)\tilde{\pi}_{1,0}(\sigma)\tilde{\pi}_{1:2,0}(\sigma)\tilde{\pi}_{0,0}(\sigma^{-1})\tilde{\pi}_{0:1,0}(\sigma^{-1})\varrho_1(0)\omega_r(1)e^{\theta_{3,4}/2}$$
$$\times (7 + 7\gamma_{2,2}^{1/2} + \gamma_{2,3}^{1/3} + \gamma_{3,2}^{1/2})\big\{1 + (\beta_{r,6n}/\alpha)^{1/6}\big\}^2.$$

Lastly, for the fourth Stein factor $\zeta_4$ using the fourth derivative of the representation (B.1) together with the bounds (B.6) and (B.7),

$$\left|\nabla^4 u_f(x)[v,u,w,y]\right| \leq \int_0^3 \left|\nabla^4(P_t f)(x)[v,u,w,y]\right| dt + \int_3^\infty \left|\nabla^4(P_t f)(x)[v,u,w,y]\right| dt$$

$$\leq \left[6\tilde{\pi}_{1:4,n}(f)\left[1 + 6\gamma_{2,4}^{1/4} + 4\gamma_{2,3} + 3\gamma_{2,3}^{2/3} + 4\gamma_{3,3}^{1/3} + \gamma_{4,2}^{1/2}\right]e^{3\theta_{4,2}/2}\left\{1 + (\beta_{r,6n}/\alpha)^{1/6}\right\}\right.$$

$$\left. + \xi_4 \int_3^\infty \varrho_1(t-3)\omega_r(t-1)dt\right](1 + \|x\|_2^n)\|u\|_2\|v\|_2\|w\|_2\|y\|_2.$$

The final result follows from taking a supremum over $u, v, w, y$:

$$\|\nabla^4 u_f(x)\|_{\mathrm{op}} \leq \zeta_4(1 + \|x\|_2^n) \quad \text{where}$$

$$\zeta_4 = 6\tilde{\pi}_{1:4,n}(f)\left[1 + 6\gamma_{2,4}^{1/4} + 4\gamma_{2,3} + 3\gamma_{2,3}^{2/3} + 4\gamma_{3,3}^{1/3} + \gamma_{4,2}^{1/2}\right]e^{3\theta_{4,2}/2}\left\{1 + (\beta_{r,6n}/\alpha)^{1/6}\right\}$$

$$+ \xi_4 \int_0^\infty \varrho_1(t)\omega_r(t+2)dt,$$

and

$$\xi_4 = 4\tilde{\mu}_{1,n}(f)\tilde{\pi}_{1,0}(\sigma)^2\tilde{\pi}_{1:3,0}(\sigma)\tilde{\pi}_{0,0}(\sigma^{-1})^2\tilde{\pi}_{0:2,0}(\sigma^{-1})e^{\theta_{4,2}}\varrho_1(0)\omega_r(1)$$

$$\times \left\{1 + (\beta_{r,6n}/\alpha)^{1/6}\right\}^3 \left[42 + 32\gamma_{2,2}^{1/2} + 6\gamma_{2,2} + 2\gamma_{2,3}^{1/3} + 3\gamma_{2,3}^{2/3} + 24\gamma_{2,4}^{1/4} + 3\gamma_{2,4}^{1/2} + 12\gamma_{2,6}^{1/6}\right.$$

$$\left. + 5\gamma_{3,2}^{1/2} + 5\gamma_{3,3}^{1/3} + \gamma_{4,2}^{1/2} + 6\gamma_{2,2}^{1/2}\gamma_{2,6}^{1/6}\right].$$

## C   Proofs of Expected Suboptimality Bounds

*Proof of Prop. 4.1.* We begin by proving the more general claim (4.2). Our dissipativity assumption together with the diffusion moment bounds in [12] implies that $p(\|\cdot\|_2^2) \leq \beta/\alpha$. Moreover, as noted in the proof of [25, Prop. 3.4], the differential entropy is bounded by that of a multivariate Gaussian with the same second moments:

$$-p(\log p) \leq \frac{d}{2}\log\left(\frac{2\pi e p(\|\cdot\|_2^2)}{d}\right). \leq \frac{d}{2}\log\left(\frac{2\pi e \beta}{d\alpha}\right).$$

Meanwhile, $\log p(x^*) = -\log \int p(x)/p(x^*)dx$. Our smoothness assumption, a polar coordinate transform, and the integral identity of [16, 3.326 2] imply that

$$\int p(x)/p(x^*)dx = \int \exp(\log p(x) - \log p(x^*))dx \geq \int \exp(-C\|x - x^*\|_2^{2\theta})dx \tag{C.1}$$

$$= \int_0^\infty S_{d-1}r^{d-1}\exp(-Cr^{2\theta})dr = S_{d-1}\frac{1}{2\theta}\Gamma\left(\frac{d}{2\theta}\right)C^{-d/(2\theta)}$$

where $S_{d-1} = 2\frac{\pi^{d/2}}{\Gamma(d/2)}$ is the surface area of the unit sphere in $\mathbb{R}^d$ and $\Gamma(\cdot)$ is the Gamma function. Since, by [18, Thm. 2], $\Gamma(x+y)/\Gamma(y) \geq x^y\frac{x}{x+y}$ for all $x, y > 0$,

$$\log p(x^*) \leq \frac{d}{2\theta}\log(C) - \log\left(\frac{S_{d-1}}{2\theta}\Gamma\left(\frac{d}{2\theta}\right)\right) = \frac{d}{2\theta}\log(C) + \frac{d}{2}\log\left(\frac{1}{\pi}\right) - \log\left(\frac{1}{\theta}\frac{\Gamma\left(\frac{d}{2\theta}\right)}{\Gamma\left(\frac{d}{2}\right)}\right) \tag{C.2}$$

$$\leq \frac{d}{2\theta}\log(C) + \frac{d}{2}\log\left(\frac{1}{\pi}\right) - \left(\frac{1}{\theta} - 1\right)\frac{d}{2}\log\left(\frac{d}{2}\right)$$

$$\leq \frac{d}{2\theta}\log\left(\frac{2C}{d}\right) + \frac{d}{2}\log\left(\frac{d}{2\pi}\right).$$

The result (4.2) now follows by summing the estimates (C.1) and (C.2).

Now consider the case in which $p = p_{\gamma,\theta}$. By design, $x^*$ is also a global minimizer of $f$ with $\nabla f(x^*) = 0$. Therefore, by Taylor's theorem, we have for each $x$

$$\log p_{\gamma,\theta}(x^*) - \log p_{\gamma,\theta}(x) = \gamma(f(x) - f(x^*))^\theta$$

$$= \gamma(\langle\nabla f(x^*), x - x^*\rangle + \tfrac{1}{2}\langle x - x^*, \nabla^2 f(z)(x - x^*)\rangle)^\theta$$

$$\leq \frac{\gamma\mu_2^\theta(f)}{2^\theta}\|x - x^*\|_2^{2\theta}.$$

The generalized Gibbs result (4.1) now follows from the general claim (4.2) and Jensen's inequality as $p_{\gamma,\theta}(\gamma(f(x) - f(x^*))^\theta) \geq \gamma p_{\gamma,\theta}^\theta(f(x) - f(x^*))$ for $\theta \in (0,1]$. $\qquad\square$

*Proof of Prop. 4.3.* Let $\alpha = 1/k$. We have

$$\mathbb{E}_{x \sim p_{\gamma,\alpha}}[f(x)] - f^* = \frac{\int((x-b)^\top A(x-b))\exp\left(-\gamma\left((x-b)^\top A(x-b)\right)^\alpha\right)dx}{\int \exp\left(-\gamma((x-b)^\top A(x-b))^\alpha\right)dx}.$$

Using the variable change $y = A^{1/2}(x-b), dy = \det(A^{1/2})dx$, the above equals

$$\frac{\int \|y\|^{2\alpha}\exp(-\gamma\|y\|^{2\alpha})dy}{\int \exp(-\gamma\|y\|^{2\alpha})dy} \;=\; \frac{\int_0^\infty S_{d-1}r^{d-1}\cdot r^2\exp(-\gamma r^{2\alpha})dr}{\int_0^\infty S_{d-1}r^{d-1}\exp(-\gamma r^{2\alpha})dr} \;=\; \frac{\int_0^\infty r^{d+1}\exp(-\gamma r^{2\alpha})dr}{\int_0^\infty r^{d-1}\exp(-\gamma r^{2\alpha})dr}\,,$$

where $S_{d-1}$ is the surface area of the unit sphere in $\mathbb{R}^d$. Substituting an explicit expression for these integrals we get

$$\frac{\Gamma\left(\frac{d+2}{2\alpha}\right)/2\alpha\gamma^{(d+2)/2\alpha}}{\Gamma\left(\frac{d}{2\alpha}\right)/2\alpha\gamma^{d/2\alpha}} \;=\; \frac{\Gamma\left(\frac{d}{2\alpha}+\frac{1}{\alpha}\right)}{\Gamma\left(\frac{d}{2\alpha}\right)}\gamma^{-1/\alpha}\,,$$

where $\Gamma(\cdot)$ is the Gamma function. Substituting back $k = 1/\alpha$, and noting that $\Gamma(z+1) = z\Gamma(z)$ for all $z$, we get that the above equals

$$\frac{\Gamma\left(\frac{dk}{2}+k\right)}{\Gamma\left(\frac{dk}{2}\right)}\gamma^{-k} \;=\; \gamma^{-k}\prod_{i=0}^{k-1}\left(\tfrac{dk}{2}+i\right) \;\leq\; \gamma^{-k}\left(\tfrac{dk}{2}+k-1\right)^k \;=\; \left(\frac{k\left(\frac{1}{2}d+1\right)-1}{\gamma}\right)^k.$$

$\square$

# D  Proof of Prop. 3.5: User-friendly Wasserstein decay for Gibbs measures

Define $\tilde{\sigma}_\gamma(x) = (\sigma_\gamma(x)\sigma_\gamma(x)^\top - s^2 I)^{1/2} = \frac{1}{\sqrt{\gamma}}\tilde{\sigma}(x)$. Our assumptions imply

$$\frac{\langle b_\gamma(x)-b_\gamma(y), x-y\rangle}{s^2\|x-y\|_2^2/2} + \frac{\|\tilde{\sigma}_\gamma(x)-\tilde{\sigma}_\gamma(y)\|_F^2}{s^2\|x-y\|_2^2} - \frac{\|(\tilde{\sigma}_\gamma(x)-\tilde{\sigma}_\gamma(y))^\top(x-y)\|_2^2}{s^2\|x-y\|_2^4}$$

$$\leq \frac{-\gamma\langle m(x)\nabla f(x)-m(y)\nabla f(y), x-y\rangle}{s_0^2\|x-y\|_2^2} + \frac{\langle\langle\nabla, m(x)-m(y)\rangle, x-y\rangle + \|\tilde{\sigma}(x)-\tilde{\sigma}(y)\|_F^2}{s_0^2\|x-y\|_2^2}$$

$$\leq \begin{cases} -\frac{\gamma K_m - L^*}{s_0^2} & \text{if } \|x-y\|_2 > R \\ \frac{\gamma L_m + L^*}{s_0^2} & \text{if } \|x-y\|_2 \leq R, \end{cases}$$

as advertised.

# E  Auxiliary Lemmas

**Lemma E.1** (Quadratic form moment bounds). *For $W_m \sim \mathsf{N}_d(0, I)$ which is independent from $X_m$, we have*

$$\mathbb{E}\big[\|\sigma(X_m)W_m\|_2^{2n}\big] \leq (2n-1)!!\,\mathbb{E}\big[\|\sigma(X_m)\|_F^{2n}\big].$$

*Proof.* The exact expressions for the quadratic form moments can be found in [21]. We simply use the properties of Frobenius norm to obtain a compact upper bound. $\square$

**Lemma E.2.** *For a sequence of real nonnegative numbers $\{a_i\}_{i=0}^n$ satisfying $a_{i+1} \leq \tau a_i + \gamma$ for $\tau \in (0,1)$ and $\gamma \in \mathbb{R}$ we have*

$$\tfrac{1}{n}\textstyle\sum_{i=1}^n a_i \leq a_0 + \tfrac{\gamma}{1-\tau}.$$

*Proof.* By the recursive inequality, we have

$$a_i \leq \tau^i a_0 + \gamma\tfrac{1-\tau^i}{1-\tau}.$$

Averaging over $i$, we obtain

$$\tfrac{1}{n}\textstyle\sum_{i=1}^n a_i \leq \tfrac{1}{n}\textstyle\sum_{i=1}^n\left(a_0\tau^i + \gamma\tfrac{1-\tau^i}{1-\tau}\right),$$

$$\leq \tfrac{a_0\tau}{n}\tfrac{1-\tau^m}{1-\tau} + \tfrac{\gamma}{1-\tau} \leq a_0 + \tfrac{\gamma}{1-\tau},$$

where in the last step, we used $\tau \leq 1$ and the Bernoulli inequality

$$1 - \tau^n = 1 - (1-(1-\tau))^n \leq n(1-\tau).$$

$\square$

**Lemma E.3.** *For $x, a, c > 0$ and $m \geq 1$, we have $ax^m + a(c/a)^m/m \geq cx^{m-1}$.*

*Proof.* The derivative of the polynomial $p(x) = ax^m - cx^{m-1} + b$ has $m - 2$ roots at 0, and a root at $x_0 = c(m-1)/(am)$. Therefore, $p(x)$ for $x \geq 0$, attains its minimum value at $x_0$. We choose $b = a(c/a)^m/m$ so that

$$
\begin{aligned}
p(x_0) &= (ax_0 - c)x_0^{m-1} + b \\
&= b - \frac{ac^m}{ma^m}\left(1 - \frac{1}{m}\right)^{m-1} \geq 0,
\end{aligned}
$$

where for the last step, we use $f(x) = (1 - 1/x)^{x-1} \leq 1$ for $x \geq 1$ and $\lim_{x \downarrow 1} f(x) = 1$. □