[Reviews · NeurIPS 2018]

Reviewer 1



This paper deals with non-convex optimization problem using Ito diffusion processes. More specifically the paper consider the problem of minimizing a function f on R^d (smooth enough). To treat this problem, the analysis of the authors is as follows. 1) They first consider the problem of sampling from the density proportional to exp(-\gamma f). Indeed using an existing result the mean of this distribution is close to the minimum of f for \gamma \to \infty. To sample from exp(-\gamma f), they consider an Euler discretization (X_k) of an Itô diffusion for which this density is stationary. They give some bounds on the bias of some empirical distribution and the target measure which are based on a Poisson solution. 2) They give some conditions to ensure that the Poisson solutions satisfied the given assumption. 3) They apply their result for minimization of functions which appear in learning problems and give some indications on appropriate choices of Itô diffusion to deal with this problem. I think that it is a nice contribution but that the main message of this paper which is about really non-convex optimization is lost because on the really technical section 3. In this section, they give a quantitative results on the weak errors estimates given in Mattingly et al depending on the Poisson solution they considered and give really nasty and non explicit bounds on the constant associated with the Poisson solution. I would be really happy if the authors really made some conclusions about these new bounds exept that the complexity is of order O(\eps^{-2}) which was known. In addition, I do not really understand what is the role of the integer n_e in the results of this section. I really like section 4 which would have required I think more explanation and details in particular the examples could be presented more deeply. In my opinion it is the main contribution of the paper. Finally it is a little unfortunate that the authors do not tries their methodology on some simple examples and real ones.

Reviewer 2



This paper studies nonconvex optimization problem using Euler discretization on Ito diffusions, and derives the convergence rate of the investigated discretized diffusions. Specifically, the authors apply Poisson equation to analyze the convergence of discretized diffusions in terms of expected function value and prove a convergence rate of O(1/\epsilon^2) under several seemingly stringent assumptions. In addition, the authors also give some examples that are not covered by existing Langevin theory but can be illustrated by the theory in this paper. I notice that the authors mentioned one recent work in the literature, i.e., [26], which is highly related to this submission. However, I didn’t find a careful comparison between this submission and [26] in the related work or the main theory section (Section 3). Since the O(1/epsilon^2) convergence results seems worse than the O(1/\epsilon) rate proved in [26],yet this work makes more assumptions beyond the smooth and dissipativity assumptions, it is very important to discuss the connection and difference between this submission and [26] thoroughly. The strengths of this paper are given as follows: 1. This paper studies a Euler discretization on general Ito diffusions, which is beyond the framework of conventional Langevin dynamics. 2. This paper presents some examples to showcase the applications of the main theory. The weakness of this paper: 1. In [22], the convergence rate of SGLD contains a spectral gap which is exponential in dimension d and inverse temperature \gamma. Similarly, the result for GLD (the full gradient version) in [26] has also such a dependence on d and \gamma. Furthermore, there is a lower bound for the metastable exiting time of SDE for nonconvex functions in Bovier et al., (2004), which has exponential dependence on d and \gamma too. However, this paper seems to hide many factors including the spectral gap in the main theory. The authors should explicitly indicate these dependences and make a fair and thorough comparison with previous work. 2. The results for nonconvex optimization is claimed to be \mathcal{O}(1/\epsilon^2), while the complexity in [26] is \tilde(O)(1/\epsilon) for the same algorithm (GLD). The authors should explain why their result looks inferior to the existing work. 3. In Section 5, the authors just simply combine the argument in Theorem 3.1 and Proposition 3.4 in [22]. Some important discussions are missing, such as the optimal choice of the inverse temperature parameter \gamma, and how it affects the convergence rate. I believe the choice of \gamma will affect both the ergodicity and discretization results. Besides, the authors attempt to improve the expected suboptimality using non-Gibbs distributions. However, a simple quadratic example is not convincing enough, and the impacts of using non-Gibbs distributions on the ergodicity were not well explained. 4. The convergence rate highly depends on the parameter n, I would like to see some discussions on this parameter (use some examples or provide an upper bound). 5. In condition 3, the authors mention \tilde \mu_n(u_f) = \zeta_1. I noticed that the notation \tilde \mu_n(u_f) has appeared before in (2.6), are they the same? 6. A typo in Line 72: it is the inverse of spectral gap, i.e., \lambda^{-1} that is exponential in d and \gamma. 7. The organization can be improved, the authors are better to clearly state the assumptions on objective function, rather than write them with plain text. Reference: Bovier, A., Eckhoff, M., Gayrard, V. and Klein, M. (2004). Metastability in reversible diffusion processes i: Sharp asymptotics for capacities and exit times. Journal of the European Mathematical Society 6 399–424. ===== After I read the authors' response, my concern on the comparison with [26] was addressed. Please incorporate the comparison with [26] in your response into the camera ready.

Reviewer 3



This paper studies non-convex optimization using general diffusion processes, with forward Euler discretization. As in Raginsky et al., the global convergence to global minimum is based on ergodic properties of the Markov process, and convergence to stationary. The tools for analysis are quite different, however, which makes it possible for this paper to study diffusion process with general diffusion matrices. The ergodic properties are characterized by Stein factor, i.e., regularity of solution to Poisson equation, up to order 4. Under certain growth conditions for the first fourth order derivatives of the objective function and the Poisson equation regularity, explicit bounds for the optimization error are established. Following previous works such as Gotham et al., the Poisson equation regularity is implied by Wasserstein decay. The techniques are not completely new: Poisson equation results are based on Mattingly et al., the reflection coupling for Wasserstein decay is from Eberle et al., and the construction of general diffusions are from Ma et al. However, it is still very interesting to see the Poisson equation techniques being applied to this field. And the author also provide concrete examples to show the effectiveness of varying diffusion matrices. The paper shows Wasserstein decay for different classes of functions, including functions with non-convex tails, which have not been covered in the literature. The varying diffusion matrix allows the ergodicity to still hold true. Detailed comments: 1. In Theorem 3.1, fourth order smoothness for the Poisson equation is used, but is it actually necessary? What will happen if we do the Taylor expansion with fewer orders? 2. The estimates for V_t^{x,v}, U_t^{x,vu}, etc. blow up exponentially with time in Appendix B. And they're used in the semigroup derivative estimates in Appendix C (e.g. Equation C.8-C.11) Why is it possible for the final estimates to avoid diverging with $T$? After author feedback: my questions are addressed in the author feedback. I like this paper and this has not changed.